# HETEROGENEOUS PERSONALIZED FEDERATED LEARNING BY LOCAL-GLOBAL UPDATES MIXING VIA CONVERGENCE RATE

**Meirui Jiang**
Department of Computer Science and Engineering
The Chinese University of Hong Kong
`mrjiang@cse.cuhk.edu.hk`

**Anjie Le**
Department of Computer Science and Engineering
The Chinese University of Hong Kong
`ajle@cuhk.edu.hk`

**Xiaoxiao Li**
Department of Electrical and Computer Engineering
The University of British Columbia
`xiaoxiao.li@ece.ubc.ca`

**Qi Dou**[*]
Department of Computer Science and Engineering
The Chinese University of Hong Kong
`qidou@cuhk.edu.hk`

## ABSTRACT

Personalized federated learning (PFL) has emerged as a promising technique for addressing the challenge of data heterogeneity. While recent studies have made notable progress in mitigating heterogeneity associated with label distributions, the issue of effectively handling feature heterogeneity remains an open question. In this paper, we propose a personalization approach by **L**ocal-**G**lobal updates **Mix**ing (LG-Mix) via Neural Tangent Kernel (NTK)-based convergence. The core idea is to leverage the convergence rate induced by NTK to quantify the importance of local and global updates, and subsequently mix these updates based on their importance. Specifically, we find the trace of the NTK matrix can manifest the convergence rate, and propose an efficient and effective approximation to calculate the trace of a feature matrix instead of the NTK matrix. Such approximation significantly reduces the cost of computing NTK, and the feature matrix explicitly considers the heterogeneous features among samples. We have theoretically analyzed the convergence of our method in the over-parameterize regime, and experimentally evaluated our method on five datasets. These datasets present heterogeneous data features in natural and medical images. With comprehensive comparison to existing state-of-the-art approaches, our LG-Mix has consistently outperformed them across all datasets (largest accuracy improvement of 5.01%), demonstrating the outstanding efficacy of our method for model personalization. Code is available at `https://github.com/med-air/HeteroPFL`.

## 1 INTRODUCTION

Personalized federated learning (PFL) aims to leverage the aggregated knowledge from other clients to learn a client-specific model that best fits its own data distribution (Smith et al., 2018; Arivazhagan et al., 2019; Li et al., 2021b; Hanzely & Richtárik, 2021; Tan et al., 2022). Although many PFL methods have tackled heterogeneity issues regarding label distribution shift (Jeong & Hwang, 2022; Zhang et al., 2022), computation limitation (Diao et al., 2021; Setayesh et al., 2023), model architecture design (Makhija et al., 2022; Wang et al., 2022), etc, how to effectively address the feature heterogeneity issue (Kairouz et al., 2021; Hsieh et al., 2020) is still an open question. In practice, feature distributions across client data are often heterogeneous due to variations in acquisition or generation conditions (Xu et al., 2021; Rieke et al., 2020; Li et al., 2021c). For instance, in healthcare domain, medical images collected from different hospitals exhibit appearance misalignment as imaging protocol changes. Training one common global model for tackling various feature distribu-

---

[*]Corresponding author.

tions can be challenging. Therefore, personalization of the global FL model to each individual client becomes imperative, but how to solve the feature heterogeneity is still unclear.

There have been many approaches proposed to improve the global model in order to tackle heterogeneous features. Despite the promising performance, the converged global model may not be optimal for all clients (Kairouz et al., 2021; Li et al., 2020a; Sattler et al., 2020). Another way is to train personalized models to overcome the feature distribution shifts. It is worth noting that dealing with heterogeneous features under PFL setting is largely different from doing it under standard FL. Specifically, PFL aims to fit clients' individual feature distribution, whereas the standard FL focuses on utilizing various feature distributions to enhance the global model. For instance, there are some FL methods proposed to tackle the heterogeneous data by aligning the feature distribution over clients to promote global aggregation (Li et al., 2021c;a), and rectifying client gradient direction to avoid aggregated gradients being distracted (Gao et al., 2022; Karimireddy et al., 2020). These methods mainly focus on global updates. However, for PFL, it is essential to consider both local and global updates and their interplay relations for achieving personalization.

To investigate the relations between local and global updates, we consider their effects during federated training. The global update typically is the aggregation (e.g., weighted averaging) of local updates, while the local update solely relies on the client's local data. In federated training, the global update contains common knowledge from all clients' data, thereby aiding in minimizing the error of joint data distribution. However, it may not precisely align with the direction of the local update when the data exhibits heterogeneity. For local update, it specifically minimizes the local error, but suffers from limited local data size. From a personalization standpoint, the key incentive is to minimize the local error with the help of other clients' data (i.e., the common knowledge). In this case, an ideal solution would be leveraging local updates to rectify the potential adverse effects induced by data heterogeneity in global updates. A promising solution is to mix the local and global updates, while the question is how to determine the mixing ratio.

One related work has proposed to mix the local and global models and select the ratio based on the differences observed between the mixed model and the global model (Deng et al., 2020). However, considering that the local/global update strongly correlates with the input data, we argue that attention must be given to the data aspect in the context of feature heterogeneous PFL. To answer this question, we draw insight from the Neural Tangent Kernel (NTK) (Jacot et al., 2018; Huang et al., 2021), which leverages the dot product of input data for measuring the convergence of neural network training. By calculating the NTK matrix when employing local and global updates, we can take the NTK-based convergence rate as a guiding factor to adaptively adjust the mixing ratio.

In this paper, we propose to achieve model personalization by **L**ocal-**G**lobal updates **Mix**ing (LG-Mix). Our key contribution is to determine the mixing ratio via NTK-based convergence. Specifically, we theoretically show that the trace of NTK is an effective measurement of the convergence rate. By leveraging the convergence rate induced by NTK, we can assess the importance of local and global updates, and perform mixing based on their importance. We propose an efficient and effective approximation to calculate the trace of a feature matrix instead of the NTK matrix. This approximation not only reduces the computational cost, but also explicitly considers heterogeneous features among samples. We conduct comprehensive experiments to demonstrate the efficacy of our method, including both performance comparisons and analytical studies. We also theoretically analyzed the convergence of our method in the over-parameterized regime. Our experiments include three computer vision datasets with heterogeneous features (diverse image styles/appearances), and two real-world medical image datasets. Our method consistently outperforms state-of-the-art approaches across all datasets, showing its effectiveness in personalization.

## 2 RELATED WORKS

**Federated Learning on Heterogeneous Data.** To promote the training of a global model on heterogeneous data, various techniques have been proposed, such as regularizing local model training (Li et al., 2020b; Durmus et al., 2021; Li et al., 2021a), facilitating model optimization (Karimireddy et al., 2020; Reddi et al., 2021; Tran Dinh et al., 2021), enhancing aggregation algorithm (Wang et al., 2020; Pillutla et al., 2019), improving feature normalization (Li et al., 2021c; Reisizadeh et al., 2020), etc. For instance, SCAFFOLD (Karimireddy et al., 2020) proposed a new optimization algorithm to reduce variance in local updates. Later on, FedNova (Wang et al., 2020) suggested

using normalized stochastic gradients for global model aggregation, and MOON (Li et al., 2021a) proposed using contrastive learning on latent feature representations to enhance the alignment between local and global models. FedDC (Gao et al., 2022) improved SCAFFOLD by dynamically updating the client objective function. However, when aiming to maximize the performance for each local client, learning one common global model may not be an optimal solution.

**Personalization in Federated Learning.** The PFL aims to utilize data from multiple clients to learn a personalized model for each client. Existing methods have performed personalization by leveraging: meta-learning (Fallah et al., 2020; Acar et al., 2021), multi-task learning (Smith et al., 2018; Li et al., 2021b), model parameters decomposition (Collins et al., 2021; Oh et al., 2022), model mixture (Deng et al., 2020; Hanzely et al., 2020; Hanzely & Richtárik, 2021), Bayesian treatment (Kotelevskii et al., 2022; Ozkara et al., 2023), etc. For example, PerFedAvg (Fallah et al., 2020) proposed to seek a meta-model that adapts to each client's local dataset. APFL (Deng et al., 2020) and L2SGD (Hanzely et al., 2020) proposed to mix the local and global model for personalization. FedAlt (Pillutla et al., 2022) proposes to personalize partial model layers. FedBABU (Oh et al., 2022) and FedRep (Collins et al., 2021) propose a similar idea of decoupling the learning model into a model body and a local head, and the head is personalized. Recently, FedHKD (Chen et al., 2023) proposes to use knowledge distillation to train local models and share hyper-knowledge instead of parameters. However, although some optimization-based methods (e.g., APFL, L2SGD) can be extended for heterogeneous features, they are not specifically designed for feature heterogeneity problems. The issue of feature distributional shifts in PFL remains under-explored.

## 3 LG-MIX: LOCAL-GLOBAL UPDATES MIXING VIA CONVERNGENCE

In this section, we begin by describing the model training process involving local and global updates. Then we present our proposed method that performs update mixing guided by the convergence.

### 3.1 LOCAL AND GLOBAL UPDATES MIXING

Assume we have $N$ clients joining the PFL for $T$ communication rounds, and we use $c$ to denote the client index. Each client will perform $K$ local update steps. We denote the aggregated server model at the $t$-th communication round as $u(t)$ and the client model in round $t$ and local step $k$ as $w_c(t, k)$. Denote $S_c$ as the data index of client $c$, assuming there are $n$ input data and label pairs $\{(x_i, y_i)\} \in \mathbb{R}^d \times \mathbb{R}\}_{i=1}^n$, which follow the global distribution $D$. Note that $S_1 \cup \cdots \cup S_N = [n]$, and $S_i \cap S_j = \phi$. The data of client $c$ is $\{(x_i, y_i) : i \in S_c\}$, and it follows distribution $D_c$.

The local update is defined as the weight changes before and after local client training. Consider the local model $w_c$, the local update and global update at $t$-th round can be expressed as:

$$\Delta w_c(t) = w_c(t, K) - w_c(t, 0), \quad \Delta u(t) = u(t+1) - u(t). \tag{1}$$

Here, the global update takes the format of aggregation using local updates in practice, i.e., $\Delta u(t) = \sum_c p_c \Delta w_c(t)$, where $p_c$ denotes the client importance (e.g., proportional to client sample number) and the sum over all clients equals to 1. In FL, a typical paradigm is that each client receives the global update during communication and then use it to update the local model. However, clients may experience covariate shift or concept drift in the real world, i.e., given the joint probability $P_c(x, y)$ of input $x$ and label $y$, we have $P_c(x)$ varies even if $P_c(y|x)$ is the same or $P_c(x|y)$ varies across clients while $P_c(y)$ is unchanged. Therefore, $D$ does not necessarily represent the actual distribution of a client $c$'s data $D_c$ well. That is, the global update may not concisely align with the direction of the local update, solely using the global update is not sufficient to minimize the error of each client.

To overcome the potential shift inside the data distribution, we propose to perform the PFL by incorporating the local update into the global update to fit $D_c$. The insight is that the global update contains common knowledge from all clients and conveys global information, while the local update specifically encompasses local information. By mixing local and global updates, we can make the best use of both local and global knowledge to maximize the performance for each client. Specifically, we consider the following update regime for model personalization:

$$w_c(t+1) \leftarrow w_c(t) + \lambda_c(t)\Delta w_c(t) + (1 - \lambda_c(t))\Delta u(t), \tag{2}$$

where $\lambda_c$ is a coefficient of our choice to balance the local and global update. When $\lambda_c$ approaches 0 or 1, the update regime corresponds to the vanilla FedAvg and local gradient descent respectively. Next, we aim to find an optimal ratio to mix these two updates.

## 3.2 MIXING RATIO CALCULATION BY NTK-CONVERGENCE

To find the optimal mixing ratio, the key point is to quantify which update is more important for model optimization. In other words, the optimal mixing should minimize the loss faster than other mixing ratios. Therefore, we propose to measure the convergence of using local/global updates, and use the convergence rate to determine the mixing ratio. We measure the convergence by using the tool NTK, which has been widely adopted to analyze the convergence of modern neural networks.

To study the convergence of the gradient descent at time step $t$, we can measure the evolution of the error between the ground truth and model prediction at the $t$-th step with the help of the Gram matrix $H(t)$. The Gram matrix $H(t)$ is a function of parameter $w$ and input $x$, which characterizes the optimization process of gradient descent (Du et al., 2019). This Gram matrix is often used as an empirical approximation to the NTK, which describes the dynamics of gradient descent in the infinite wide neural network (Arora et al., 2019a; Lee et al., 2020), and the spectral property of the Gram matrix also governs convergence guarantees for networks in the finite width case (Huang & Yau, 2020; Zhang et al., 2020; Brand et al., 2021). For instance, consider a two-layer neural network with ReLU activation, the $H(t)$ for input data pair $x_i, x_j$ can be denoted as: $H(t)_{i,j} = \frac{1}{m} \sum_{r \in [m]} \left( x_i^T x_j \mathbb{1}_{w_r(t)^T x_i \geq 0, w_r(t)^T x_j \geq 0} \right)$, where $m$ is the number of hidden nodes, $r$ is the node index. We omit the footnote $c$ in $w$ for ease of notation. Intuitively, the Gram matrix captures the correlations between the training samples in the network evolution dynamics. Following Du et al. (2019), we express the evolution of prediction error for one step of gradient descent as:

$$y - y(t+1) = (I - \eta H(t))(y - y(t+1)), \tag{3}$$

where $y$ denotes the label, $y(t) = f(w(t), x)$ and $f : \mathbb{R}^d \to \mathbb{R}$ is the model function. Based on Eq.(3), we have the following observation, which illustrates the significance of between the convergence rate and the trace of the Gram matrix.

**Proposition 1.** *With the assumption that the error vector $\xi(t) := y - y(t)$ can be regarded as a random vector distributed uniformly in the space, by decomposing $H(t)$ and $\xi(t)$ into the eigenbasis of $H(t)$, we note that in gradient descent, the prediction error has an approximate convergence rate of $(1 - 2\eta \, tr(H(t))/n)$, where $\eta$ is the learning rate and is small, and $tr(H(t))$ is the trace of $H(t)$.*

Proof Sketch: Consider the eigen-decomposition $H(t) = \sum_{i=1}^{n} s_i(t) v_i(t) v_i^T(t)$, then $\xi(t) = \sum_{i=1}^{n} (v_i^T(t)\xi(t)) v_i(t)$. Then we have $\xi(t+1) = (I - \eta H(t))\xi(t) = \sum_{i=1}^{n} (1 - \eta s_i)(v_i^T(t)\xi(t)) v_i(t)$, which implies $\|\xi(t+1)\|^2 = \sum_{i=1}^{n} (1 - \eta s_i)^2 (v_i^T(t)\xi(t))^2$. Then for $\eta$ small and error vector distributed uniformly in space, we have $\|\xi(t+1)\|^2 \approx (1 - 2\eta \text{tr}(H(t))/n)\|\xi(t)\|^2$.

This proposition means that the trace of the Gram matrix can serve as an effective metric for assessing the convergence rate of the model on the data samples. Consequently, the convergence rate can be quantified by leveraging the trace of the $H$ matrix. Let us denote the Gram matrix of the model obtained through global updates as $H$, and the Gram matrix acquired through local updates as $H_c$. Intuitively, if the convergence rate using local updates surpasses that of global updates, a greater emphasis on local updates becomes necessary. According to the proposition, for the same amount of sample, if $\text{tr}(H_c) > \text{tr}(H)$, then this means the local update converges more rapidly than the global update. This proposition aligns with our intuitive understanding. Following this idea, we would like to give the update with higher convergence rate a greater weight in the mixing ratio. To be more specific, for client $c$ at $t$-th step, we can calculate the mixing ratio as:

$$\lambda_c(t) = \text{tr}(H_c(t)) \, / \, (\text{tr}(H_c(t)) + \text{tr}(H(t))) \, . \tag{4}$$

By taking the ratio back to Eq.(2), we have completed the regime to generate the personalized model.

## 3.3 ALGORITHM IMPLEMENTATION

In the preceding section, we have introduced the utilization of the trace of the Gram matrix for determining the mixing ratio. However, the computation of the Gram matrix $H$ in practical networks poses significant challenges and is often computationally infeasible (Fort et al., 2020; Mohamadi et al., 2023; Holzmüller et al., 2023). Specifically, the size of the $H$ matrix increases rapidly as the number of samples grows, making the computation impractical, particularly for large datasets. In this case, we propose to approximate the calculation of $\text{tr}(H)$ by two steps.

First, we draw upon previous studies (Seleznova et al., 2023; Kirichenko et al., 2023) that emphasize the significance of the last layer in the dynamics and performance of neural networks. In our

---

**Algorithm 1:** LG-Mix: Local-global updates mixing algorithm

---

**Input:** communication rounds $T$, number of clients $N$, local update steps $K$, client learning
rate $\eta_l$, global learning rate $\eta_g$, mixing ratio $\{\lambda_c\}_{c=1}^N$, feature matrix $\{\Sigma_c\}_{c=1}^N$, $\Sigma$.

1 **for** $t = 0, 1, \cdots, T-1$ **do**
2    **for** $c = 1, 2, \cdots, N$ ***in parallel*** **do**
3       **for** $k = 0, 1, \cdots, K-1$ **do**
4          sample a batch of data pairs $\{(x_i, y_i)\}, i \in S_c$
5          $w_c(t, k+1) \leftarrow SGD\big(w_c(t,k); \{(x_i, y_i)\}\big)$   // also output feature $h_c(x_i)$
6          $\Sigma_c += h_c(x_i)h_c(x_i)^T, \Sigma += h(x_i)h(x_i)^T$     // update feature matrix
7       $\Delta w_c(t) = w_c(t, K) - w_c(t, 0)$       // send client update to server
8    $u(t+1) = u(t) + \Delta u(t)$, where $\Delta u(t) = \sum_c p_c \Delta w_c(t)$    // global model update
9    **for** $c = 1, 2, \cdots, N$ ***in parallel*** **do**
10       $\lambda_c(t) = \text{tr}(\Sigma_c(t)) \,/\, (\text{tr}(\Sigma_c(t)) + \text{tr}(\Sigma(t)))$       // Eq.(6)
11       $w_c(t+1) \leftarrow w_c(t) + \lambda_c(t)\Delta w_c(t) + (1 - \lambda_c(t))\Delta u(t)$     // Eq.(2)

**Output:** The personalized models $\{w_c\}_{c=1}^N$, and the global model $u$.

---

approximation, we consider the evolution of the last layer. For the matrix $H$, we use $\hat{H}(t)_{i,j} := \frac{1}{m}\sum_{r \in [m]}\big(h_r(x_i)h_r(x_j)\big)$ as a approximated representation, where $h$ denotes the parameters up to the penultimate layer. This approximation is also justified by the fact that $h_r(x_i)h_r(x_j)$ is always a factor of $H_{i,j} = \langle \nabla f_w(x_i), \nabla f_w(x_i) \rangle$, irrespective of the activation function employed. Here, $w$ is the parameter of the last layer, and $f$ gives the prediction of the whole model. Notably, an existing work (Seleznova et al., 2023) has also shown that the empirical NTK with entries of last-layer features highly aligns with the original NTK in terms of training dynamics.

Second, since our objective is to compute the matrix trace, it is unnecessary to calculate the complete $\hat{H}$. By leveraging a property of linear algebra, namely, $\text{tr}\big(h(x)^T h(x)\big) = \text{tr}\big(h(x)h(x)^T\big)$, we can simplify the trace calculation as follows:

$$\text{tr}(\hat{H}(t)) = \frac{1}{m}\sum_{i \in [n]}\sum_{r \in [m]} h_r(x_i)h_r(x_i) = \frac{1}{m} \times \text{tr}(\Sigma(t)), \tag{5}$$

where $\Sigma(t) = \sum_{i=1}^n h(x_i)h(x_i)^T$ is a feature matrix in $\mathbb{R}^{m \times m}$ considering features among all samples. Note that $\hat{H}(t)$ has a size of $n \times n$, while $\Sigma$ is $m \times m$. As a result, the computational cost is significantly reduced since the number of samples is typically much larger than the last-layer feature dimension (e.g., 256 or 512).

With the aforementioned approximation, we can approximate the trace of the $H$ matrix obtained through local updates as $\text{tr}(H_c(t)) \approx \text{tr}(\Sigma_c(t))$, where $\Sigma_c(t) = \sum_{i=1}^{n_c} h_c(x_i)h_c(x_i)^T$. In the case of the global $H$ matrix, since the data is distributed and not directly accessible, we propose employing features from local data in conjunction with the global model $u$. Consequently, we approximate $\text{tr}(H(t))$ as $\text{tr}(\Sigma(t))$, where $\Sigma(t) = \sum_{i=1}^{n_c} h(x_i)h(x_i)^T$.

Finally, the mixing ratio for client $c$ at time step $t$ can be calculated as follows:

$$\lambda_c(t) = \text{tr}(\Sigma_c(t)) \,/\, (\text{tr}(\Sigma_c(t)) + \text{tr}(\Sigma(t))). \tag{6}$$

To further stabilize the ratio, we propose to consider the history information, that is, $\lambda_c(t) = \frac{1}{t-1}\sum_{i=1}^{t-1}\lambda_c(i)$. We have also studied the effects of such stabilization in our experiments.

## 4   CONVERGENCE ANALYSIS

We prove the following theorem that describes the convergence of our proposed PFL algorithm.

**Theorem 1.** *For $m = \Omega(\lambda^{-4}n^4 \log(n/\delta))$, randomly initialized parameters (i.e. $w(0) \sim \mathcal{N}(0, I)$), and $\eta_l = \mathcal{O}(\lambda/\kappa K n^2)$, $\eta_g = \mathcal{O}(1)$, then with probability at least $1 - \delta$ over the random initialization, we have for $\forall t$:*

$$\|y - y(t+1)\|^2 \leq \|y - y(t)\|^2 - \zeta\eta_g(1 - \lambda(t))s_{min}^{(H)}\|y - y(t)\|^2 - \zeta\sum_c \lambda(t)s_{min}^{(H_c)}\|y_c - y_c(t)\|^2,$$

*where $\zeta := \frac{\eta_l K}{2N}$, and $s_{min}$ denotes the smallest eigenvalue of $H$ matrix.*

Note that $\|y - y(t)\|^2 = \sum_c \|y_c - y_c(t)\|^2$, and

$$y(t) = (f(w_1(t), x_1), ..., f(w_1(t), x_{n_1}), f(w_2(t), x_1), ....., f(w_N(t), x_{n_N}))^T$$

$$y_c(t) = (0, ..., 0, \underbrace{f(w_c(t), x_1), ..., f(w_c(t), x_{n_c})}_{x_i \in S_c}, 0, ..., 0)^T$$

The proof is rather involved including bounding each of the terms in the recursion relation:

$$\|y - y(t+1)\|^2 = \|y - y(t)\|^2 - 2(y - y(t))^T (y(t+1) - y(t)) + \|y(t+1) - y(t)\|^2$$

We defer the details of the proof to Appendix B.

**Discussion on the convergence result.** To gain an intuition into the convergence result, we consider the case when $\lambda$ is not homogeneous among clients, and we consider a sub-optimal inequality as can be deduced from Appendix B:

$$\|y-y(t+1)\|^2 \leq \|y-y(t)\|^2 - \zeta\eta_g(1-\lambda_{max}(t))s_{min}^{(H)}\|y-y(t)\|^2 - \zeta\sum_c \lambda_c(t)s_{min}^{(H_c)}\|y_c-y_c(t)\|^2$$

Then by rewriting $s_{min}^{(H_c)} = s_{min}^{(H)} + \delta_c$, and $\|y_c - y_c(t)\|^2 = \frac{1+\epsilon_c}{N}\|y - y(t)\|^2$, we have

$$\|y-y(t+1)\|^2 \leq \|y-y(t)\|^2 - \zeta\|y-y(t)\|^2\Big[\eta_g(1-\lambda_{max}(t))s_{min}^{(H)} + \bar{\lambda}_c s_{min}^{(H)} + \frac{1}{N}\sum_c \lambda_c\delta_c(1+\epsilon_c)\Big]$$

where $\bar{\lambda}_c$ denotes the average value of $\{\lambda_c\}$. With the first two terms in the square bracket being relatively fixed, we observe that with $\epsilon_c = o(1)$, larger value of $\lambda_c$ is preferred for larger value of $\delta_c$.

## 5 EXPERIMENTS

### 5.1 EXPERIMENTAL SETTINGS

**Datasets.** We evaluate our approach on four classification datasets, including (1) *Digits-5* (Zhou et al., 2020; Li et al., 2021c) with digits images showing drastic differences in font style, color, and background. (2) *Office-Caltech10* (Gong et al., 2012) with images acquired in different cameras or environments; (3) *DomainNet* (Peng et al., 2019) with different image styles; (4) *Camelyon17* (Bandi et al., 2018) with histology images with different stainings from 5 hospitals, and one segmentation task on the *Retinal* dataset which contains retinal fundus images from 6 different sources (Fumero et al., 2011; Sivaswamy et al., 2015; Almazroa et al., 2018; Orlando et al., 2020). As shown in Fig. 1, each client represents a data source, and data are heterogeneous across clients.

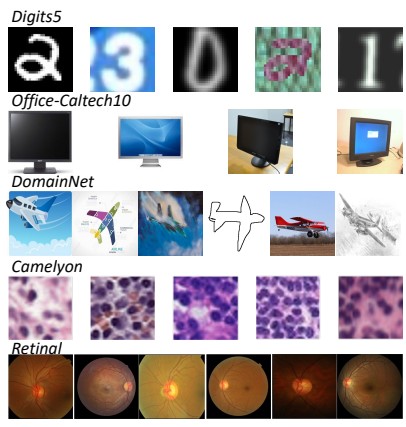

Figure 1: Samples of five datasets show various styles and appearances.

**Compared methods and evaluation metrics.** We compare our approach with state-of-the-art (SOTA) PFL methods, including APFL (Deng et al., 2020) and L2SGD (Hanzely et al., 2020) for personalization by mixing local and global models, which shares a similar idea of ours; FedAlt (Pillutla et al., 2022) for personalizing partial model layers; PerFedAvg (Fallah et al., 2020) for learning a meta-model that adapts to each client's local data; FedBN (Li et al., 2021c) for personalizing batch normalization layers; FedFomo (Zhang et al., 2021) for aggregating certain client models based on client contribution; FedBABU (Oh et al., 2022) and FedRep (Collins et al., 2021), which propose to personalize the last model layer; FedHKD (Chen et al., 2023) for using knowledge distillation to personalize local models. For evaluation metrics, we report the accuracy for all classification tasks, and the Dice coefficient (Dice) and Hausdorff Distance (HD) for the segmentation task. All results are reported with mean and standard deviation across three independent runs.

**Implementation details.** In our implementation, all methods use the same training settings. We use the SGD optimizer with a learning rate of $0.01$ and CrossEntropy loss for classification tasks and use Adam optimizer with learning rate of $1e^{-3}$ with $\beta = (0.9, 0.99)$, dice loss (Milletari et al., 2016) for the segmentation task. For more experimental results and training implementation details, please refer to Appendix. D.

Table 1: Performance comparison with SOTA PFL methods on classification datasets of Digits5 (different strokes and colors) and Office-Caltech10 (different shapes and view angles).

| Dataset | Digits5 | | | | | | Office-Caltech10 | | | | |
|---|---|---|---|---|---|---|---|---|---|---|---|
| Client | A | B | C | D | E | Avg. | A | B | C | D | Avg. |
| FedAvg (PMLR 2017) | 98.85 (0.03) | 89.95 (0.09) | 95.82 (0.48) | 99.28 (0.0) | 88.70 (0.37) | 94.52 (0.14) | 78.18 (2.36) | 56.99 (1.69) | 51.04 (6.51) | 61.58 (3.91) | 61.95 (1.13) |
| APFL (Arxiv) | 96.62 (2.47) | 90.07 (0.53) | 96.94 (0.89) | 99.12 (0.15) | 91.17 (2.31) | 94.78 (0.43) | 78.05 (3.21) | 53.48 (0.86) | 64.76 (17.04) | 57.91 (13.46) | 63.55 (0.42) |
| L2SGD (NeurIPS 2020) | 98.87 (0.05) | 89.99 (0.07) | 96.00 (0.2) | 99.29 (0.01) | 88.84 (0.37) | 94.60 (0.1) | 78.18 (2.36) | 56.99 (1.69) | 51.04 (6.51) | 68.25 (1.86) | 63.62 (1.18) |
| FedAlt (ICML 2022) | 99.20 (0.05) | 90.51 (0.27) | 98.26 (0.38) | 99.32 (0.03) | 92.36 (0.08) | 95.93 (0.08) | 77.31 (2.69) | 55.95 (2.01) | 44.79 (1.8) | 75.14 (4.27) | 63.30 (0.37) |
| PerFedAvg (NeurIPS 2020) | 99.05 (0.06) | 89.55 (0.12) | 96.11 (0.11) | 99.23 (0.02) | 89.53 (0.27) | 94.69 (0.1) | 71.73 (1.39) | 56.55 (2.2) | 61.46 (4.77) | 74.01 (3.53) | 65.94 (0.61) |
| FedBN (ICLR 2021) | 99.22 (0.16) | 91.48 (0.2) | 96.20 (0.11) | 99.32 (0.01) | 91.14 (0.58) | 95.47 (0.16) | 80.10 (0.91) | **58.18** (2.46) | 79.17 (3.61) | 83.05 (4.48) | 75.13 (1.75) |
| FedFomo (ICLR 2021) | 98.83 (0.04) | 90.49 (0.44) | 95.95 (0.22) | 99.33 (0.04) | 89.19 (0.38) | 94.76 (0.15) | 74.76 (0.9) | 54.69 (1.09) | 54.58 (2.01) | 67.34 (2.84) | 62.84 (0.69) |
| FedRep (ICML 2021) | 98.86 (0.12) | 90.35 (0.09) | 95.99 (0.47) | 99.53 (0.01) | 89.15 (0.19) | 94.78 (0.16) | 78.53 (1.05) | 57.74 (0.93) | 56.25 (5.41) | 67.23 (5.95) | 64.94 (2.76) |
| FedBABU (ICLR 2022) | 98.85 (0.04) | 90.15 (0.17) | 95.75 (0.39) | **99.53** (0.01) | 88.74 (0.49) | 94.60 (0.17) | 77.84 (0.6) | 57.74 (1.29) | 56.25 (6.25) | 67.23 (7.06) | 64.76 (2.74) |
| FedHKD (ICLR 2023) | 98.11 (0.48) | 90.38 (0.19) | 95.41 (0.88) | 99.47 (0.09) | 90.06 (0.87) | 94.69 (0.36) | 77.14 (1.31) | 56.52 (1.46) | 53.98 (0.34) | 65.48 (3.04) | 63.28 (0.72) |
| LG-Mix (Ours) | **99.29** (0.03) | **92.35** (0.17) | **98.66** (0.05) | 99.41 (0.02) | **95.77** (0.05) | **97.10** (0.04) | **80.45** (0.6) | 56.55 (0.68) | **86.46** (1.8) | **93.79** (2.59) | **79.31** (1.06) |

Table 2: Performance comparison with SOTA PFL methods on classification datasets of DomainNet (different styles) and Camelyon17 (different stainings).

| Dataset | DomainNet | | | | | | | Camelyon17 | | | | | |
|---|---|---|---|---|---|---|---|---|---|---|---|---|---|
| Client | A | B | C | D | E | F | Avg. | A | B | C | D | E | Avg. |
| FedAvg (PMLR 2017) | 58.17 (0.0) | 35.06 (1.65) | 53.61 (2.52) | 51.97 (0.99) | 67.05 (1.95) | 54.15 (0.72) | 53.34 (0.53) | 95.44 (1.05) | 92.20 (0.62) | 93.96 (0.88) | 97.21 (0.27) | 97.71 (0.33) | 95.30 (0.28) |
| APFL (Arxiv) | 58.11 (1.35) | 36.87 (2.0) | 53.29 (1.8) | 46.60 (7.54) | 69.02 (3.02) | 58.00 (0.81) | 53.65 (1.61) | 96.27 (0.29) | 92.94 (1.23) | 96.70 (0.4) | 97.95 (0.13) | 98.25 (0.15) | 96.42 (0.23) |
| L2SGD (NeurIPS 2020) | 59.00 (0.72) | 34.60 (2.13) | 53.88 (1.28) | 50.33 (2.73) | 69.82 (2.33) | 56.02 (1.67) | 53.94 (0.26) | 96.67 (0.33) | 93.02 (0.32) | 94.71 (0.18) | 97.55 (0.15) | 97.68 (0.26) | 95.93 (0.13) |
| FedAlt (ICML 2022) | 59.82 (0.48) | 37.14 (0.97) | 58.47 (1.47) | 58.57 (3.65) | 72.45 (2.0) | 54.27 (0.73) | 56.79 (0.69) | 98.10 (0.18) | 95.51 (0.07) | 98.41 (0.01) | 98.80 (0.01) | 98.74 (0.11) | 97.91 (0.06) |
| PerFedAvg (NeurIPS 2020) | 59.57 (1.52) | 35.42 (0.99) | 55.99 (1.06) | 48.47 (0.31) | 67.60 (0.92) | 56.08 (2.64) | 53.85 (0.39) | 96.71 (0.52) | 93.05 (0.41) | 95.06 (0.47) | 97.68 (0.4) | 97.92 (0.2) | 96.08 (0.3) |
| FedBN (ICLR 2021) | 58.17 (0.87) | 36.94 (1.14) | 55.61 (1.75) | 69.10 (1.91) | 73.25 (3.61) | 53.31 (4.68) | 57.73 (0.69) | 96.65 (0.49) | 92.84 (0.45) | 94.22 (0.4) | 97.55 (0.13) | 97.60 (0.31) | 95.77 (0.21) |
| FedFomo (ICLR 2021) | 59.28 (2.37) | 36.38 (1.07) | 56.43 (1.58) | 49.40 (4.8) | 69.33 (2.19) | 57.34 (1.09) | 54.69 (1.35) | 96.67 (0.34) | 92.25 (0.51) | 95.18 (0.16) | 97.60 (0.16) | 96.42 (0.47) | 95.62 (0.13) |
| FedRep (ICML 2021) | 60.08 (3.67) | 36.33 (0.98) | 56.36 (0.76) | 47.80 (4.51) | 67.98 (2.05) | 58.78 (0.91) | 54.56 (0.95) | 97.07 (0.15) | 93.64 (0.37) | 96.79 (0.05) | 98.12 (0.15) | 98.28 (0.1) | 96.78 (0.08) |
| FedBABU (ICLR 2022) | 60.71 (3.17) | 37.14 (0.72) | 56.26 (1.65) | 44.63 (1.03) | 68.50 (3.06) | 59.12 (1.31) | 54.40 (0.75) | 96.69 (0.06) | 92.93 (0.53) | 94.30 (0.66) | 97.53 (0.19) | 97.41 (0.2) | 95.77 (0.2) |
| FedHKD (ICLR 2023) | 59.04 (0.46) | 36.78 (1.37) | 54.01 (1.25) | 48.81 (1.29) | 67.82 (1.35) | 56.84 (1.29) | 53.88 (0.41) | 96.32 (0.3) | 93.91 (0.76) | 94.75 (0.62) | 96.91 (0.72) | 97.56 (0.26) | 95.89 (0.24) |
| LG-Mix (Ours) | **60.84** (0.38) | **37.20** (0.95) | **61.49** (0.74) | **78.07** (1.04) | **78.62** (0.67) | **60.23** (1.15) | **62.74** (0.12) | **98.77** (0.02) | **97.98** (0.09) | **98.93** (0.07) | **99.13** (0.09) | **98.95** (0.04) | **98.75** (0.05) |

## 5.2 PERFORMACNE COMPARISON RESULTS

We first present the classification performance on four datasets, which cover the heterogeneous features regarding covariate shift and concept drift. Table 1 and 2 report all results, including each client and the average performance. It can be observed that most PFL methods outperform the common global model learned by FedAvg. As FedBN is specifically designed for heterogeneous features, it presents larger improvements than other methods on most tasks. Interestingly, we find the FedAlt, which personalizes partial layers (i.e., the output layer), also clearly outperforms other PFL methods on most tasks. This may be owing to its alternating update strategy and the importance of the output layer in classification tasks. Compared with all methods, our approach shows significant improvements on 18 over 20 clients on four datasets, with the largest average performance improvements of 5.01% and least average performance improvements of 0.84%. This demonstrates the effectiveness of our strategy specifically designed by considering the data heterogeneity.

We further perform the comparison on the retinal fundus image segmentation. The fundus image varies with different machines, illumination conditions, field of view, etc. The results are shown in Table. 3. Note that we do not include FedBABU, FedRep, and FedHKD because they are specifically designed for classification and suffer a significant performance drop on segmentation. The data

Table 3: Comparison with SOTA PFL methods on the real-world retinal fundus image segmentation.

| | | | | | | | | | | | | | | |
|---|---|---|---|---|---|---|---|---|---|---|---|---|---|---|
| | Retinal Fundus Image Segmentation | | | | | | | | | | | | | |
| Client | A | B | C | D | E | F | Avg. | A | B | C | D | E | F | Avg. |
| | Dice Coefficient (Dice) ↑ | | | | | | | Hausdorff Distance (HD) ↓ | | | | | | |
| FedAvg (PMLR 2017) | 83.95 (1.73) | 83.00 (1.05) | 81.15 (2.1) | 87.98 (0.51) | 67.60 (0.66) | 91.07 (0.32) | 82.46 (0.12) | 7.71 (0.42) | 4.77 (0.02) | 8.35 (2.59) | 4.19 (0.9) | 68.14 (31.41) | 2.38 (2.38) | 15.92 (5.04) |
| APFL (Arxiv) | 74.03 (10.16) | 72.76 (9.94) | 70.03 (10.41) | 81.30 (6.63) | 66.27 (3.93) | 90.56 (0.75) | 75.83 (6.55) | 27.91 (18.2) | 52.47 (0.1) | 43.07 (27.87) | 18.00 (15.41) | 76.69 (42.1) | 5.73 (5.73) | 37.31 (22.22) |
| L2SGD (NeurIPS 2020) | 84.46 (0.42) | 83.73 (0.29) | 82.20 (1.57) | 88.40 (0.19) | 68.90 (0.66) | **91.12** (0.28) | 83.14 (0.39) | 7.55 (0.11) | 4.81 (0.02) | 7.95 (2.59) | 4.18 (1.5) | 57.76 (15.55) | 2.48 (2.48) | 14.12 (2.84) |
| FedAlt (ICML 2022) | 85.01 (2.03) | 85.19 (0.91) | 84.06 (1.93) | 88.98 (0.67) | 64.48 (2.46) | 91.05 (0.36) | 83.13 (0.57) | 6.20 (0.39) | 5.07 (0.02) | 5.25 (0.54) | 3.59 (0.3) | 75.60 (43.05) | 2.58 (2.58) | 16.38 (7.22) |
| PerFedAvg (NeurIPS 2020) | 85.87 (0.4) | 84.55 (0.75) | 84.74 (1.53) | 88.75 (0.38) | 67.91 (1.77) | 91.10 (0.12) | 83.82 (0.29) | 7.32 (0.23) | 4.54 (0.02) | 7.79 (4.21) | 3.82 (0.61) | 73.58 (39.54) | 2.47 (2.47) | 16.59 (6.04) |
| FedBN (ICLR 2021) | 84.77 (0.29) | 83.26 (0.51) | 83.88 (0.61) | 88.45 (0.5) | 67.03 (1.55) | 91.07 (0.09) | 83.08 (0.4) | 7.60 (0.25) | 4.82 (0.01) | 5.69 (0.4) | 2.95 (0.15) | 63.34 (18.31) | 2.70 (2.7) | 14.52 (2.99) |
| FedFomo (ICLR 2021) | 71.48 (5.94) | 80.47 (0.75) | 76.62 (5.67) | 86.19 (1.01) | 55.10 (2.3) | 89.87 (0.71) | 76.62 (2.66) | 12.29 (2.37) | 4.71 (0.06) | 12.08 (3.07) | 5.10 (0.4) | 154.01 (25.43) | 2.72 (2.72) | 31.82 (3.47) |
| LG-Mix (Ours) | **89.25** (0.54) | **86.76** (0.31) | **85.86** (0.67) | **89.79** (0.08) | **83.95** (1.11) | 90.86 (0.07) | **87.75** (0.17) | **4.43** (0.25) | **3.63** (0.01) | **4.53** (0.15) | **3.57** (0.09) | **6.38** (1.2) | **2.34** (2.34) | **4.15** (0.22) |

from client E (fifth column of Retinal in Fig. 1) shows different appearances due to its different image settings (dual) from others (mono). This scenario further illustrates the necessity of model personalization. Our approach consistently outperforms all compared methods. Specifically, for client E with a unique imaging setting, our method shows very large improvements (15.05%) on Dice compared with the second best, while most PFL methods fail to present a high performance.

## 5.3 ANALYTICAL STUDIES

We further analyzed the key properties of our method, including (a) the personalization weight change during training, (b) why our personalization is helpful, (c) the client scalability of our method, (d) the distance between the final personalized model and global model, and (e) the effects of considering history personalization weights.

**Trend of the personalization weight.** We compare our personalization weight with APFL, which finds the optimal mixing ratio based on differences between mixed and global models. We present the weight curves on the Digits5 dataset in Figure 2 (a). APFL tends to mix local and global models evenly (in the range of 0.5 - 0.6). In contrast, our method tends to use more local updates at the beginning and gradually decreases the personalization weight. It finally leads to a stable value. This trend fits the intuition that the global update might be distracted by heterogeneous features and may not be very informative at the early training stage. Client D is always higher than 0.6, we speculate that client D has more data samples than others, which contribute most to global updates.

**Feature value distribution.** We further study the feature value distribution. This helps further validate the quality of the learned personalized model. If a model learns confident representations, then the related neurons should be highly activated Zhou et al. (2018), i.e., the values are higher than the activation criteria (0 for ReLU in our analysis). The results on five clients from the Digits5 dataset are shown in Figure 2 (b). Our analysis shows that our personalized model achieves significantly higher feature values than learning a FedAvg global model on heterogeneous features. Moreover, our method also outperforms client standalone training on some clients, such as clients A, B, and E. This demonstrates the effectiveness of our method in learning accurate and effective representations by incorporating both local and global knowledge.

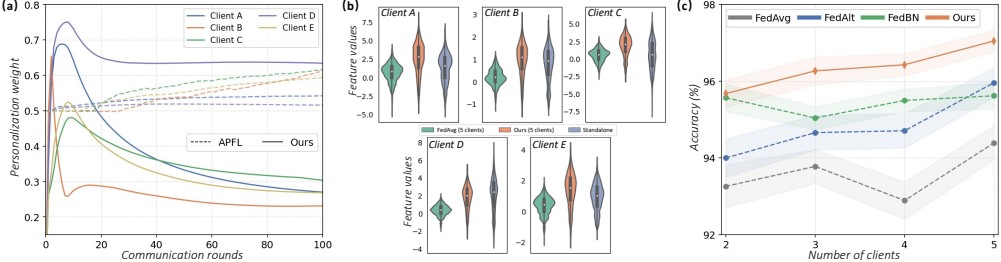

Figure 2: Analytical studies on key features of our method, including (a) the personalization weight changes; (b) the benefits of personalization via observing feature values; (c) client scalability.

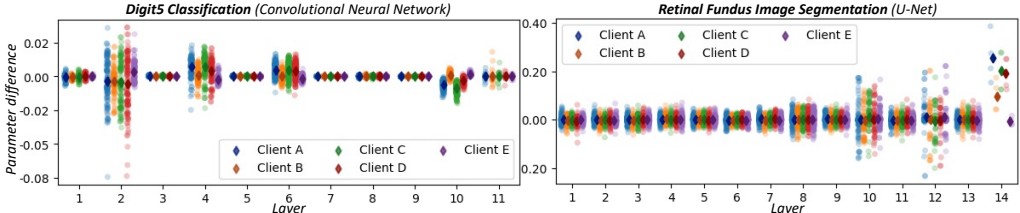

Figure 3: Distance between local and global model on the classification and segmentation model.

**Client scalability study.** We perform the client scalability analysis on our proposed method by comparing it with the two better-performing methods, FedAlt and FedBN, on the Digits5 dataset. We report the average test accuracy by increasing the number of training clients, which also results in different training feature distributions. The results are shown in Figure 2 (c), it can be observed that all PFL methods show better scalability than the baseline FedAvg. In particular, our method shows a stable increasing trend with a lower standard deviation among clients (shaded area), while performance improvements of FedBN are saturated by involving more clients, and FedAlt presents an unstable trend with a larger standard deviation.

**Distance between personalized and global model.** We also analyze the distance between the personalized model and the global model, in order to investigate how updates mixing reflect in model parameters. Specifically, we report the differences (i.e., $w_c - u$ in layer-wise) of each layer in the classification/segmentation model in Figure 3. Our analysis shows that the output layer varies significantly from the global model in classification, which validates the design of personalizing the classification head in FedBABU, FedRep, and FedAlt. Note that layers 2,4,6,8 are batch normalization layers, which also supports the idea of FedBN. However, our method does not specify any target layers but effectively personalizes these important layers identified by other methods. As for the segmentation model, all layers show variance, and the differences become larger in the decoder and output layer. This explains the failure of methods like FedBABU and FedRep, which only personalized the output layer. Overall, our mixing of local and global updates can effectively personalize the proper model parameters across all layers.

**Ablation study on using history personalization weights.** We conduct ablation studies to evaluate the effectiveness of our strategy for

Table 4: Ablation study for effects of stabilizing personalization weights by considering the history weights.

| Task | Digits5 | Office-Caltech10 | DomainNet | Camelyon17 | Retinal (Dice) |
|---|---|---|---|---|---|
| Second best | 95.93 (0.08) | 75.13(1.75) | 57.73 (0.69) | 97.91 (0.06) | 83.82(0.29) |
| Ours w/o history | 97.10 (0.05) | 78.25 (1.12) | 60.33(2.23) | 98.70(0.13) | 87.73 (0.26) |
| Ours | **97.10** (0.04) | **79.31**(1.06) | **62.74**(0.12) | **98.75**(0.05) | **87.75**(0.17) |

stabilizing personalization weights. We compare our method with the second-best method on each dataset and report the results in Table. 4. The results show that utilizing historical personalization weights improves performance on all classification tasks and has lower standard deviation, which validates the effectiveness of our stabilization strategy.

## 6 CONCLUSION

In this work, we proposed a novel approach to address the challenge in PFL for heterogeneous features. Our approach mixes local and global updates by measuring the NTK-based convergence during training. Specifically, we take the trace of NTKs using local/global updates as the hint to perform the mixing, and approximate the NTK calculation as a feature matrix calculation for computation efficiency. Besides the empirical solutions and significant performance improvements, we also theoretically analyze the convergence rate of our method using NTK. Our approach has no strict restrictions on model architectures and can be applied to a wide range of PFL applications. For future work, it is promising to investigate the effectiveness of our method on other data heterogeneity, such as the class distributional shift, and the larger model, such as the transformer.

**Acknowledgement.** This work was supported in part by National Natural Science Foundation of China (Project No. 62201485), in part by Hong Kong Research Grants Council Project No. T45-401/22-N, in part by Science, Technology and Innovation Commission of Shenzhen Municipality Project No. SGDX20220530111201008, in part by Canada NSERC Discovery Grant (RGPIN-2022-05316).

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

## APPENDIX A    NOTATION

Table 5: Major notations occurred in the paper.

| Notations | Dimension | Description |
|---|---|---|
| $k, K$ | $\mathbb{N}$ | number and total number of local update steps |
| $c, N$ | $\mathbb{N}$ | client index, total number of clients |
| $n_c, n$ | $\mathbb{N}$ | number of samples of client $c$ and overall |
| $m$ | $\mathbb{N}$ | number of hidden neurons |
| $\eta_l, \eta_g$ | $\mathbb{R}$ | client-level learning rate, global learning rate |
| $w_c, u$ | $\mathbb{R}^p$ | local model of client $c$, global model |
| $(x_i, y_i)$ | $\mathbb{R}^m$ | sample pair of index $i$ |
| $y_c(t), y(t)$ | $\mathbb{R}$ | model prediction at time $t$ with local model, global model |
| $S_c$ | set of size $n_c$ | collection of samples of client $c$ |

## APPENDIX B    GLOBAL CONVERGENCE

### B.1    FORMULATION

#### B.1.1    THE NEURAL NETWORK

For the following theoretical analysis, we follow Du et al. (2019)Song & Yang (2020)Huang et al. (2021) to consider an one-hidden-layer neural network with ReLU activation:

$$f(u, x) = \frac{1}{\sqrt{m}} \sum_{r=1}^{m} a_r \phi(u_r^T x)$$

where $m$ is the total number of neurons in the hidden layer, $a_r$'s are Rademacher random variables (take values $\{\pm 1\}$ with equal probability) and $\phi$ is the activation function. Each client aim to optimize its MSE loss:

$$L_c^{mse}(u, x) = \frac{1}{2} \sum_{i \in S_c} \left( f(u, x_i) - y_i \right)^2$$

where $S_c$ represents the collection of data of client $c$. The global loss is taken as the average of the loss of each client:

$$L(u, x) = \frac{1}{N} \sum_{c \in [N]} L_c(u, x)$$

#### B.1.2    OUR ALGORITHM

We first formulate our proposed algorithm mathematically to establish consistent notations and facilitate analysis.

In FL, each client alternate between performing local updates on its local data and communicating with the central server for global aggregation. Each client takes $K$ local updates between each communication round. The local update is performed using vanilla gradient descent with a local learning rate $\eta_l$, and $w_c(t, k)$ represents the weight parameters of client $c$ at global round $t$ and local step $k$:

$$w_c(t, k + 1) \leftarrow w_c(t, k) - \eta_l \frac{\partial L_c(w_c(t, k))}{\partial w_c(t, k)}$$

After each communication, the global aggregation procedure is conducted by taking the average of local updates of all $N$ clients, and a learning rate of $\eta_g$ is added for the global update:

$$\Delta u(t) = \frac{\eta_g}{N} \sum_{c \in [N]} \Delta w_c(t)$$

where $\Delta w_c(t) = w_c(t, K) - w_c(t, 0) = -\sum_k \eta_l \frac{\partial L_c(w_c(t,k))}{\partial w_c(t,k)}$ is the cumulative local updates of client $c$ at global round $t$. For the local step, a combination of local update and global update is taken, with $\lambda(t) \in [0, 1]$ being the combination factor:

$$w_c(t + 1, 0) \leftarrow w_c(t, 0) + (1 - \lambda(t))\Delta u(t) + \lambda(t)\Delta w_c(t)$$

### B.1.3 GRADIENT UPDATES

With the above setting, we can explicitly write out the gradient updates:

$$\Delta w_{c,r}(t) = -\sum_{k\in[K]} \eta_l \frac{\partial L_c^{mse}(w_{c,r}(t,k))}{\partial w_{c,r}(t,k)} = -\frac{\eta_l}{\sqrt{m}} \sum_{k\in[K]} \sum_{i\in S_c} [f(w_{c,r}, x_i) - y_i] a_r x_i \mathbb{1}_{w_{c,r}^T x_i \geq 0}$$

$$\Delta u_r(t) = -\frac{\eta_l \eta_g}{N\sqrt{m}} \sum_{c\in[N]} \sum_{k\in[K]} \sum_{i\in S_c} [f(w_{c,r}(t,k), x_i) - y_i] a_r x_i \mathbb{1}_{w_{c,r}^T x_i \geq 0}$$

## B.2 CONVERGENCE ANALYSIS

We analyze the convergence behavior of all clients collectively. That is, we consider the dynamics of $\|y - y(t)\|^2 = \sum_c \|y_c - y_c(t)\|^2$, where

$$y(t) = (f(w_1(t), x_1), ..., f(w_1(t), x_{n_1}), f(w_2(t), x_1), ....., f(w_N(t), x_{n_N}))^T$$

$$y_c(t) = (0, ..., 0, \underbrace{f(w_c(t), x_1), ..., f(w_c(t), x_{n_c})}_{x_i \in S_c}, 0, ..., 0)^T$$

are the stacked vector of predictions and $y, y_c$ are the corresponding ground truth.

Note first the following recurrence relation (†):

$$\begin{aligned} \|y - y(t+1)\|^2 &= \|[y - y(t)] - [y(t+1) - y(t)]\|^2 \\ &= \|y - y(t)\|^2 - 2(y - y(t))^T(y(t+1) - y(t)) + \|y(t+1) - y(t)\|^2 \\ &= \|y - y(t)\|^2 - 2\underbrace{\sum_c (y_c - y_c(t))^T(y_c(t+1) - y_c(t))}_{\text{the cross term}} + \|y(t+1) - y(t)\|^2 \end{aligned}$$

We will express $\|y - y(t+1)\|^2$ in terms of $\|y - y(t)\|^2$ with a shrinking factor, by bounding each of these terms, and hence prove the convergence of the algorithm.

### B.2.1 THE CROSS TERM

We first investigate the cross term. Note that the difficulty in the analysis mainly comes from the non-linear activation pattern. However, this is overcame by a key observation in classical NTK theoryDu et al. (2019)Huang et al. (2021) that the activation patterns stay the same for most of the neurons.

We follow their approaches to define

$$Q_i := \{r \in [m] : \forall m \in \mathbb{R}^d s.t. \|w - w_r(0)\|_2 \leq R, \mathbb{1}_{w_r(0)^T x_i \geq 0} = \mathbb{1}_{w^T x_i \geq 0}\}$$

which represent the set of neurons whose activation pattern does not change during training for sample $x_i$, and let $\bar{Q}_i$ denote its complement. Then for each sample $i \in S_c$,

$$\begin{aligned} y_i(t+1) - y_i(t) &= \frac{1}{\sqrt{m}} \sum_{r\in[m]} a_r \Big[\phi(w_r^T(t+1)x_i) - \phi(w_r^T(t)x_i)\Big] \\ &= \underbrace{\frac{1}{\sqrt{m}} \sum_{r\in Q_i} a_r(1 - \lambda_c(t))\Delta u_r^T(t)x_i \mathbb{1}_{w_r^T(t)x_i \geq 0}}_{v_{1,i}} \\ &\quad + \underbrace{\frac{1}{\sqrt{m}} \sum_{r\in Q_i} a_r \lambda_c(t)\Delta w_{c,r}^T(t)x_i \mathbb{1}_{w_r^T(t)x_i \geq 0}}_{v_{2,i}} + v_{3,i} \end{aligned}$$

where

$$v_{1,i} = -\frac{(1 - \lambda_c(t))\eta_l \eta_g}{Nm} \sum_{k \in [K], r \in Q_i} \sum_{j \in S'_c, c' \in [N]} (y(t,k)_j - y_j) x_i^T x_j \mathbb{1}_{w_{c',r}^T(t,k)x_j \geq 0, w_{c,r}^T(t)x_i \geq 0}$$

$$v_{2,i} = -\frac{\lambda_c(t)\eta_l}{m} \sum_{k \in [K], r \in Q_i} \sum_{j \in S_c} (y_c(t,k)_j - y_{c,j}) x_i^T x_j \mathbb{1}_{w_{c,r}^T(t,k)x_j \geq 0, w_r^T(t)x_i \geq 0}$$

$$v_{3,i} = \frac{1}{\sqrt{m}} \sum_{r \notin Q_i} a_r \Big[ \phi(w_r^T(t+1)x_i) - \phi(w_r^T(t)x_i) \Big]$$

We can already notice the almost symmetric kernel factor in the terms above. We give the formal definitions here.

**Definition 1** (Global Gram matrix). *For $t \in [T], k \in [K], c, c' \in [N], i \in S_c$ and $j \in S_{c'}$, we define the global gram matrix as:*

$$H(t,k)_{i,j} := \frac{1}{m} \sum_{r \in [m]} x_i^T x_j \mathbb{1}_{w_{c,r}^T(t)x_i \geq 0, w_{c',r}^T(t,k)x_j \geq 0} \qquad \in \mathbb{R}^{n \times n}$$

$$H(t,k)_{i,j}^{\perp} := \frac{1}{m} \sum_{r \notin Q_i} x_i^T x_j \mathbb{1}_{w_{c,r}^T(t)x_i \geq 0, w_{c',r}^T(t,k)x_j \geq 0} \qquad \in \mathbb{R}^{n \times n}$$

Note that this definition is similar to, but not exactly the same as, the definition in FL-NTKHuang et al. (2021). This is because they considered the vanilla FedAvg with no personalization of model parameters.

**Definition 2** (Local Gram matrix). *For $t \in [T], k \in [K], c \in [N]$ and $i, j \in S_{c'}$, we define the local gram matrix as:*

$$H_c(t,k)_{i,j} = \frac{1}{m} \sum_r x_i^T x_j \mathbb{1}_{w_{c,r}^T(t)x_i \geq 0, w_{c,r}^T(t,k)x_j \geq 0} \qquad \in \mathbb{R}^{n_c \times n_c}$$

$$H_c(t,k)_{i,j}^{\perp} = \frac{1}{m} \sum_{r \notin Q_i} x_i^T x_j \mathbb{1}_{w_{c,r}^T(t)x_i \geq 0, w_{c,r}^T(t,k)x_j \geq 0} \qquad \in \mathbb{R}^{n_c \times n_c}$$

However, in order to maintain consistent dimensions and correspond to our definition of $y_c$, we can extend the dimension of $H_c$ to $n \times n$ by adding zeros to the undefined entries, i.e.,

$$\begin{pmatrix} 0 & \cdots & 0 \\ \vdots & H_c & \vdots \\ 0 & \cdots & 0 \end{pmatrix} \in \mathbb{R}^{n \times n}$$

From now on, the symbol $H_c$ will refer to this $n \times n$ matrix. Note that $(H_c)_{i,j} = H_{i,j} \mathbb{1}_{i,j \in S_c}$.

We will show that the convergence can be governed by the spectral property of these Gram matrices. Substitute them into the cross term, we get:

$$\sum_c (y_c - y_c(t))^T (y_c(t+1) - y_c(t)))$$

$$= \sum_c \frac{(1 - \lambda_c)\eta_l \eta_g}{N} \sum_{i \in S_c} (y_{c,i} - y_c(t)_i) \sum_{k \in [K], j \in [n]} (y(t,k)_j - y_j)(H(t,k)_{i,j} - H(t,k)_{i,j}^{\perp})$$

$$+ \sum_c \lambda_c \eta_l \sum_{i \in S_c} (y_{c,i} - y_c(t)_i) \sum_{k \in [K], j \in S_c} (y_c(t,k)_j - y_{c,j})(H_c(t,k)_{i,j} - H_c(t,k)_{i,j}^{\perp})$$

$$- \sum_c \sum_{i \in S_c} (y_{c,i} - y_c(t)_i) v_{3,i}$$

Let

$$C_1 := -\sum_c \frac{(1-\lambda_c(t))\eta_l\eta_g}{N} \sum_{i \in S_c} (y_{c,i} - y_c(t)_i) \sum_{k,j} (y(t,k)_j - y_j)(H(t,k)_{i,j} - H(t,k)_{i,j}^\perp)$$

$$C_2^{(c)} := -\lambda_c(t)\eta_l \sum_{i \in S_c} (y_{c,i} - y_c(t)_i) \sum_{k,j \in S_c} (y_c(t,k)_j - y_{c,j})(H_c(t,k)_{i,j} - H_c(t,k)_{i,j}^\perp)$$

$$C_2 := \sum_c C_2^{(c)}$$

$$C_3 := -\sum_c \sum_{i \in S_c} (y_{c,i} - y_c(t)_i)v_{3,i}$$

Then by substituting them back into the recursive relation (†), we get:

$$\|y - y(t+1)\|^2 = \|y - y(t)\|^2 + 2(C_1 + C_2 + C_3) + \|y(t+1) - y(t)\|^2$$

We will bound each of these terms and hence prove the result.

## B.3 Convergence analysis - Main Theorem

We first restate the main convergence theorems.

**Theorem 1.** *For uniform* $\lambda_c(t) = \lambda(t), \forall c$, *for* $m = \Omega(\lambda^{-4}n^4 \log(n/\delta))$, *randomly initialized parameters (i.e.* $w(0) \sim \mathcal{N}(0, I)$), *and* $\eta_l = \mathcal{O}(\lambda/\kappa K n^2)$, $\eta_g = \mathcal{O}(1)$, *then with probability at least* $1 - \delta$ *over the random initialization, we have for* $\forall t$:

$$\|y - y(t+1)\|^2 \le \|y - y(t)\|^2 - \zeta\eta_g(1 - \lambda(t))s_{min}^{(H)}\|y - y(t)\|^2 - \zeta \sum_c \lambda(t)s_{min}^{(H_c)}\|y_c - y_c(t)\|^2$$

*where* $\zeta := \frac{\eta_l K}{2N}$.

**Theorem 2.** *For non-uniform* $\lambda_c(t)$, *let* $\lambda_{min}(t) := \min_c \lambda_c(t)$ *and* $\lambda_{max}(t) := \max_c \lambda_c(t)$, *For* $m = \Omega(\lambda^{-4}n^4 \log(n/\delta))$, *randomly initialized parameters (i.e.* $w(0) \sim \mathcal{N}(0, I)$), *and* $\eta_l = \mathcal{O}(\lambda/\kappa K n^2)$, $\eta_g = \mathcal{O}(1)$, *then with probability at least* $1 - \delta$ *over the random initialization, we have for* $\forall t$:

$$\|y - y(t+1)\|^2 \le \|y - y(t)\|^2 - \zeta\eta_g(1 - \lambda_{max}(t))s_{min}^{(H)}K\|y - y(t)\|^2 - \zeta \sum_c \lambda_{min}(t)s_{min}^{(H_c)}\|y_c - y_c(t)\|^2$$

*where* $\zeta := \frac{\eta_l K}{2N}$.

We will give the proof of theorem 1 in the subsequent sections, and we note that theorem 2 is a natural extension of theorem 1 so the proof also naturally extends.

## B.4 Useful Lemmas

Before giving the proof of the theorem, we state two useful lemmas.

The first lemma gives bounds on the norm of the local and global updates.

**Lemma 1.** *With* $\|x_i\|_2 = 1$, *we have*

$$\|\Delta u_r(t)\|_2 \le \frac{2\eta_l\eta_g K(1 + 2\eta_l n K)\sqrt{n}}{N\sqrt{m}}\|y - y(t)\|_2$$

$$\|\Delta w_r^{(c)}(t)\|_2 \le \frac{2\eta_l K(1 + 2\eta_l n_c K)\sqrt{n_c}}{\sqrt{m}}\|y_c - y_c(t)\|_2$$

*Proof.* The first inequality follows from FL-NTK. For the second inequality, consider:

$$\|\Delta w_r^{(c)}(t)\|_2 = \eta_l \left\| \frac{a_r}{\sqrt{m}} \sum_{k \in [K]} \sum_{i \in S_c} [y(t,k)_i - y_i] x_i \mathbb{1}_{w_{k,c}^T x_i \geq 0} \right\|$$

$$\leq \frac{\eta_l}{\sqrt{m}} \sum_{k \in [K]} \sum_{i \in S_c} |y_i - y(t,k)_i|$$

$$\leq \frac{\eta_l \sqrt{n_c}}{\sqrt{m}} \sum_{k \in [K]} \|y_c - y_c(t,k)\|$$

$$\leq \frac{\eta_l K (1 + 2\eta_l n_c K) \sqrt{n_c}}{\sqrt{m}} \|y_c - y_c(t)\|_2$$

$\square$

The second lemma bounds the sum of client prediction error by that of the global error.

**Lemma 2.**

$$\sum_c \|y_c - y_c(t)\| \leq \sqrt{N} \|y - y(t)\|$$

*Proof.* By Jensen's inequality, and since the square root function is concave,

$$\sqrt{\frac{1}{N} \sum_c \|y_c - y_c(t)\|^2} \geq \frac{1}{N} \sum_c \|y_c - y_c(t)\|$$

$\square$

### B.5 PROOF OF THEOREM 1

We provide here a detailed proof of theorem 1, and we note that the same proof naturally extends to prove theorem 2. We will use $\lambda$ to represent $\lambda(t)$ for ease of notation.

Firstly, here are two results that directly follow from FL-NTK. They provide bounds on the effect of global and local updates respectively.

**Proposition 2.** *With probability at least $1 - n \exp(-mR)$ over random initialization, we have*

$$C_1 \leq \frac{\eta_l \eta_g (1 - \lambda)}{N} \|y - y(t)\|^2 (-K s_{min}^{(H)} + 40 \sqrt{n} R K (1 + 2\eta_l K \sqrt{n}) + 2\eta_l s_{max}^{(H)} K^2 \sqrt{n}))$$

$$+ \frac{8\eta_g \eta_l (1 - \lambda)}{N} K (1 + 2\eta_l n K) n R \|y - y(t)\|^2$$

**Proposition 3.** *With probability at least $1 - n \exp(-mR)$ over random initialization, we have*

$$C_2^{(c)} \leq \frac{\lambda \eta_l}{N} \|y_c - y_c(t)\|^2 (-K s_{min}^{(H_c)} + 40 \sqrt{n} R K (1 + 2\eta_l K \sqrt{n} + 2\eta_l s_{max}^{(H_c)} K^2 \sqrt{n}))$$

$$+ \frac{8\lambda \eta_l}{N} K (1 + 2\eta_l n K) n R \|y_c - y_c(t)\|^2$$

For the following two propositions, we assume that all clients possess the same number of samples, i.e., $n_c = n/N, \forall c$. Additionally, let $\tilde{\eta}_g$ denote $\max\{1, \eta_g\}$.

The following proposition aims to bound the effect of updates on neurons whose activation pattern changed during the algorithm.

**Proposition 4.** *With probability at least $1 - n \exp(-mR)$ over random initialization, we have*

$$C_3 \leq \frac{8\eta_l \tilde{\eta}_g K}{N} (1 + 2\eta_l n K) n R \|y - y(t)\|^2$$

*Proof.* Consider

$$\|v_3\|_2^2 \le \underbrace{\frac{1-\lambda}{m} \sum_{i\in[n]} \Big(\sum_{r\in\bar{Q}_i} |\Delta u_r(t)^T x_i|\Big)^2}_{A} + \underbrace{\frac{\lambda}{m} \sum_{i\in[n]} \Big(\sum_{r\in\bar{Q}_i} |\Delta w_r(t)^T x_i|\Big)^2}_{B}$$

and

$$A \le \Big(\frac{8(1-\lambda)\eta_g \eta_l K}{N}(1+2\eta_l nK)nR\|y-y(t)\|\Big)^2$$

As for $B$,

$$
\begin{aligned}
B &= \frac{\lambda}{m} \sum_c \sum_{i\in S_c} \Big(\sum_{r\in[m]} \mathbb{1}_{r\in\bar{Q}_i} |\Delta u_r(t)^T x_i|\Big)^2 \\
&\le \frac{\lambda\eta_l^2}{m} \sum_c \frac{4K^2(1+2\eta_l n_c K)^2 n_c}{m}\|y_c - y_c(t)\|^2 \cdot n_c(4mR)^2 \\
&\le \Big(\frac{8\lambda\eta_l K}{N}(1+2\eta_l nK)nR\|y-y(t)\|\Big)^2
\end{aligned}
$$

where for the second inequality we used the assumption made above. Then

$$
\begin{aligned}
C_3 &:= -\sum_{i\in[n]} (y_i - y_i(t))v_{3,i} \\
&\le \|y-y(t)\|_2 \|v_3\|_2 \\
&\le \frac{8\eta_l \tilde{\eta}_g K}{N}(1+2\eta_l nK)nR\|y-y(t)\|^2
\end{aligned}
$$

$\square$

Now we have bounded the cross term. For the last term, we have the following inequality:

**Proposition 5.** *We have*

$$\|y(t+1)-y(t)\|^2 \le \frac{4\eta_l^2 \tilde{\eta}_g^2 n^2 K^2 (1+2\eta_l nK)^2}{N^2}\|y-y(t)\|^2$$

*Proof.*

$$
\begin{aligned}
\|y(t+1)-y(t)\|^2 &\le \frac{1-\lambda}{m} \sum_{i\in[n]} \Big(\sum_{r\in[m]} |\Delta u_r(t)^T x_i|\Big)^2 + \frac{\lambda}{m} \sum_{i\in[n]} \Big(\sum_{r\in[m]} |\Delta w_r(t)^T x_i|\Big)^2 \\
&\le \frac{(1-\lambda)\eta_g^2 \eta_l^2}{m} \Big(\frac{2K(1+2\eta_l nK)\sqrt{n}}{N\sqrt{m}}\|y-y(t)\|\Big)^2 \cdot nm^2 \\
&\quad + \sum_c \frac{\lambda\eta_l^2}{m} \Big(\frac{2K(1+2\eta_l n_c K)\sqrt{n_c}}{\sqrt{m}}\|y_c - y_c(t)\|\Big)^2 \cdot n_c m^2 \\
&\le \frac{4\eta_l^2 \tilde{\eta}_g^2 n^2 K^2 (1+2\eta_l nK)^2}{N^2}\|y-y(t)\|^2
\end{aligned}
$$

where we have used the assumption that $n_c = n/N$. $\square$

Now by substituting the above results to the recursion equation, we get:

$$
\begin{aligned}
\|y - y(t+1)\|^2 \leq \|y - y(t)\|^2 \\
+ \frac{2\eta_l \eta_g (1-\lambda)}{N} \|y - y(t)\|^2 (-K s_{min}^{(H)} + 40\sqrt{n} RK(1 + 2\eta_l K\sqrt{n}) \\
+ 2\eta_l s_{max}^{(H)} K^2 \sqrt{n})) + \frac{16\eta_g \eta_l (1-\lambda)}{N} K(1 + 2\eta_l nK)nR\|y - y(t)\|^2 \\
+ \sum_c \frac{2\lambda \eta_l}{N} \|y_c - y_c(t)\|^2 (-K s_{min}^{(H_c)} + 40\sqrt{n} RK(1 + 2\eta_l K\sqrt{n} \\
+ 2\eta_l s_{max}^{(H_c)} K^2 \sqrt{n})) + \frac{16\lambda \eta_l}{N} K(1 + 2\eta_l nK)nR \sum_c \|y_c - y_c(t)\|^2 \\
+ \frac{16\eta_l \tilde{\eta}_g K}{N}(1 + 2\eta_l nK)nR\|y - y(t)\|^2 \\
+ \frac{4\eta_l^2 \tilde{\eta}_g^2 n^2 K^2 (1 + 2\eta_l nK)^2}{N^2} \|y - y(t)\|^2
\end{aligned}
$$

Then by the choice of $\eta_l \leq \min\{\frac{s_{min}^{(H)}}{1000\kappa n^2 K}, \min_c\{\frac{s_{min}^{(H_c)}}{1000\kappa_c n^2 K}\}\}$ where $\kappa := s_{max}/s_{min}$ and $\eta_l \eta_g \leq \min\{\frac{s_{min}^{(H)}}{1000\kappa n^2 K}, \min_c\{\frac{s_{min}^{(H_c)}}{1000\kappa_c n^2 K}\}\}$ and $R \leq s_{min}^{(H)}/(1000n)$, we have

$$
\begin{aligned}
\|y - y(t+1)\|^2 \leq \|y - y(t)\|^2 \\
- \frac{(1-\lambda)\eta_l \eta_g s_{min}^{(H)} K}{N} \|y - y(t)\|^2 - \sum_c \frac{\lambda \eta_l s_{min}^{(H_c)} K}{N} \|y_c - y_c(t)\|^2 \\
+ 40\frac{\eta_l \eta_g KnR}{N} \|y - y(t)\|^2 \times 2 \\
+ \frac{\eta_l^2 \tilde{\eta}_g^2 n^2 K^2}{N^2} \|y - y(t)\|^2 \\
\leq \|y - y(t)\|^2 - \frac{(1-\lambda(t))\eta_l \eta_g s_{min}^{(H)} K}{2N} \|y - y(t)\|^2 \\
- \sum_c \frac{\lambda(t)\eta_l s_{min}^{(H_c)} K}{2N} \|y_c - y_c(t)\|^2
\end{aligned}
$$

by substituting in the condition on $\eta_l$ and $R$.

Quod erat demonstrandum.

## APPENDIX C   GENERALIZATION

In this section, we prove the generalization bounds. That is, we aim to find a bound on

$$
\mathcal{L}_{\mathcal{D}}(f) := \mathbb{E}_{(x,y)\sim D}[l(f(x), y)]
$$

where $f$ refer to the prediction function we consider. Note that, in practice, this is approximated by the empirical loss $L_S(f) = \frac{1}{n} \sum_{i\in[n]} l(f(x_i), y_i)$. We also consider a more general initialization scheme $w_r \sim \mathcal{N}(0, \sigma^2 I)$.

### C.1   SETUP

We follow Arora et al. (2019b); Huang et al. (2021) to consider a non-degenerate data distribution.

**Definition 3** (Non-degenerate Data Distribution). *A distribution $\mathcal{D}$ over $\mathbb{R}^b \times \mathbb{R}$ is $(\lambda, \delta, n)$-non-degenerate, if with probability at least $1-\delta$, for $n$ iid samples $\{(x_i, y_i)\}_{i=1}^n$ chosen from $\mathcal{D}$, $s_{\min}^{(H^\infty)} \geq s > 0$.*

We also state here the definition of the dynamic matrices which can be used to describe the evolution of the neural network:

**Definition 4** (Global Trajectory Matrix).

$$
J(t,k) = \frac{1}{\sqrt{m}}
\begin{pmatrix}
a_1 x_1 \mathbb{1}_{w_{c_1,1}^T(t,k)x_1 \geq 0} & \cdots & a_1 x_n \mathbb{1}_{w_{c_n,1}^T(t,k)x_n \geq 0} \\
\vdots & \ddots & \vdots \\
a_m x_1 \mathbb{1}_{w_{c_1,m}^T(t,k)x_1 \geq 0} & \cdots & a_m x_n \mathbb{1}_{w_{c_n,m}^T(t,k)x_n \geq 0}
\end{pmatrix}
\in \mathbb{R}^{md \times n}
$$

**Definition 5** (Local Trajectory Matrix).

$$
J_c(t,k) = \frac{1}{\sqrt{m}}
\begin{pmatrix}
a_1 x_1 \mathbb{1}_{w_{c_1,1}^T(t,k)x_1 \geq 0} & \cdots & a_1 x_{n_c} \mathbb{1}_{w_{c_{n_c},1}^T(t,k)x_{n_c} \geq 0} \\
\vdots & \ddots & \vdots \\
a_m x_1 \mathbb{1}_{w_{c_1,m}^T(t,k)x_1 \geq 0} & \cdots & a_m x_{n_c} \mathbb{1}_{w_{c_{n_c},m}^T(t,k)x_{n_c} \geq 0}
\end{pmatrix}
\in \mathbb{R}^{md \times n_c}
$$

*for $x_i$'s sample of client c, and where appropriate, we fill in the undefined entries with $0$ to form a matrix of dimension $md \times n$.*

Note that $H = J^T J$ and $H_c = J_c^T J_c$. We also give some useful notations following the above definitions.

**Notation 1.**
$$
\tilde{J}(t,k) = (J_{c_1}(t,k), J_{c_2}(t,k), \cdots, J_{c_N}(t,k)) \in \mathbb{R}^{md \times n}
$$

**Notation 2.**
$$
\tilde{H} =
\begin{pmatrix}
H_1 & \cdots & 0 \\
\vdots & \ddots & \vdots \\
0 & \cdots & H_N
\end{pmatrix}
\in \mathbb{R}^{n \times n}
$$

We also use a notation $vec(A)$ to express the vectorization of a matrix $A$ in column-first order. Then the gradient update rule can be expressed as:

$$
\begin{aligned}
vec(W_c(t,k+1)) &= vec(W_c(t,k)) - \eta_l J_c(t,k)(y_c(t,k) - y_c) \\
vec(U(t+1)) &= vec(U(t)) - \frac{\eta_l \eta_g}{N} \sum_k J(t,k)(y(t,k) - y) \\
vec(W_c(t+1)) &= vec(W_c(t)) - \lambda \eta_l \sum_k J_c(t,k)(y_c(t,k) - y_c) \\
&\quad - (1-\lambda)\frac{\eta_l \eta_g}{N} \sum_k J(t,k)(y(t,k) - y)
\end{aligned}
\tag{7}
$$

## C.2 SOME USEFUL RESULTS

We first quote a result from Huang et al. (2021) which will be used later.

**Lemma 3.** *For $J(t,k)$ as defined above, with probability at least $1 - n\exp(-m\exp(-m(R\sigma^{-1} + \delta)/10))$, we have*
$$
\|J(t,k) - J(0,0)\|_F \leq 2n(R\sigma^{-1} + \delta)
$$

The following lemma give an approximation on the dynamics of the global model.

**Lemma 4.** *For $A(\lambda) = (1-\lambda)\frac{\eta_l \eta_g K}{N}H^\infty + \lambda \eta_l K \tilde{H}_c^\infty$ and $\beta(\lambda) = (1-\lambda)\frac{\eta_l \eta_g K}{N} + \lambda \eta_l K$, we have*
$$
y(t) - y = -(I - A(\lambda))^t y + e(t)
$$

*where*
$$
\|e(t)\|_2 \leq \mathcal{O}\left((1 - \beta(\lambda)s_{min})^t \left(\sqrt{n}\sigma + \frac{t\beta(\lambda)n^{7/2}}{s_{min}\sigma\sqrt{m}}\right) poly(\log(m/\delta))\right)
$$

*Proof.* Recall that from Appendix B, we have $[y(t) - y] - [y(t-1) - y] = v_1 + v_2 + v_3$, and that

$$v_{1,i} = -\frac{(1-\lambda)\eta_l\eta_g K}{N} \sum_{j \in [n]} (y_j(t) - y_j) H^\infty_{i,j}$$

$$-\frac{(1-\lambda)\eta_l\eta_g}{N} \sum_{j \in [n],k} (y_j(t,k) - y_j(t)) H^\infty_{i,j}$$

$$-\frac{(1-\lambda)\eta_l\eta_g}{N} \sum_{j \in [n],k} (y_j(t,k) - y_j)(H(t,k)_{i,j} - H^\infty_{i,j})$$

$$-\frac{(1-\lambda)\eta_l\eta_g}{N} \sum_{j \in [n],k} (y_j(t,k) - y_j)(H^\perp(t,k)_{i,j})$$

and similar for $v_{2,i}$ except that $i, j \in S_c$ for some client $c$.

Let

$$\xi_i(t) := v_{1,i}(t) + v_{2,i}(t) + v_{3,i}(t)$$

$$+ \frac{(1-\lambda)\eta_l\eta_g K}{N} \sum_{j \in [n]} (y_j(t) - y_j) H^\infty_{i,j}$$

$$+ \lambda\eta_l K \sum_{j \in S_c} (y_j(t) - y_j)(H^\infty_c)_{i,j}$$

Note that by Appendix B, $\|v_3(t)\| = \frac{16\eta_l\tilde{\eta}_g K}{N}(1 + 2\eta_l nK)nR\|y - y(t)\|$, $\|y_c(t) - y_c(t,k)\| \le 2\eta_l nK\|y_c(t) - y_c\|$, $\|y - y(t,k)\|_2 \le 2(1 + 2\eta_l nK)\|y - y(t)\|_2$, $\|H(w,w) - H(w_1,w_2)\|_F \le 4nR$ and $\|H(t,k)^\perp\|_F \le 4nR$ etc. By taking the maximum order among the terms, we have that

$$\|\xi(t)\|_2 \le \mathcal{O}\Big(\frac{\beta(\lambda)n^3 s_{max}\sqrt{\log(m\delta)\log^2(n/\delta)}}{\sigma\lambda\sqrt{m}}\|y - y(t)\|_2\Big)$$

where $\beta(\lambda) := \frac{(1-\lambda)\eta_l\eta_g K}{N} + \lambda\eta_l K$.

Then

$$y(t) - y = (I - A(\lambda))(y(t-1) - y) + \xi(t-1)$$

$$= (I - A(\lambda))^t(y(0) - y) + \sum_{\tau \in [t-1]} (I - A(\lambda))^\tau \xi(t - 1 - \tau)$$

$$= -(I - A(\lambda))^t y + e(t)$$

where

$$e(t) = (I - A(\lambda))^t y(0) + \sum_{\tau \in [t-1]} (I - A(\lambda))^\tau \xi(t - 1 - \tau)$$

and since

$$\|y(0)\|_2^2 \le n\sigma^2 \cdot 2\log(2mn/\delta) \cdot \log^2(4n/\delta)$$

we have

$$\|e(t)\|_2$$

$$\le \mathcal{O}\Big((1 - \beta(\lambda)s_{min})^t\Big(\sqrt{n\sigma^2}\sqrt{2\log(2mn/\delta)}\log(8n/\delta) + t\frac{\beta(\lambda)n^{7/2}\log(m/\delta)\log^2(n/\delta)}{s_{min}\sigma\sqrt{m}}\Big)\Big)$$

$$\le \mathcal{O}\Big((1 - \beta(\lambda)s_{min})^t\Big(\sqrt{n}\sigma + \frac{t\beta(\lambda)n^{7/2}}{s_{min}\sigma\sqrt{m}}\Big)\text{poly}(\log(m/\delta))\Big)$$

$\square$

### C.3 AVERAGE GENERALIZATION

When examing a new OOD sample, we would use the average of all current parameters for prediction. Therefore, we first examine the generalization performance of the average of all paramters:

$$W(t) := \frac{1}{N} \sum_c W_c(t)$$

By (7), we have:

$$vec(W(t+1)) = vec(W(t)) - \frac{\lambda \eta_l}{N} \sum_{k,c} J_c(t,k)(y_c(t,k) - y_c) - (1-\lambda)\frac{\eta_l \eta_g}{N} \sum_k J(t,k)(y(t,k) - y)$$

**Lemma 5.** *For $A(\lambda) = (1-\lambda)\frac{\eta_l \eta_g K}{N} H^\infty + \lambda \eta_l K \tilde{H}_c^\infty$ and $\gamma(\lambda) = (1-\lambda)\frac{\eta_l \eta_g K}{N} + \lambda \frac{\eta_l K}{N}$, we have*

$$\|W(t) - W(0)\|_F \le (y^T A(\lambda)^{-T} H^\infty A(\lambda)^{-1} y)^{1/2}$$
$$+ \mathcal{O}\Big(\frac{n\sigma}{s_{min}} \cdot poly(\log(m/\delta)) + \frac{n^4}{\sigma^{1/2}m^{1/4}} \cdot poly(\log(m/\delta))\Big)$$

*Proof.*

$$vec(W(T)) - vec(W(0))$$

$$= \sum_{t \in [T-1]} \Big[ -(1-\lambda)\frac{\eta_l \eta_g}{N} \sum_k J(t,k)(y(t,k) - y) - \frac{\lambda \eta_l}{N} \sum_c \sum_k J_c(t,k)(y_c(t,k) - y_c) \Big]$$

$$= \sum_{t \in [T-1],k} -\frac{\gamma(\lambda)}{K} J(t,k)(y(t,k) - y)$$

$$= \sum_{t \in [T-1],k} \frac{\gamma(\lambda)}{K} J(t,k)(I - A(\lambda))^t y - \sum_{t \in [T-1],k} \frac{\gamma(\lambda)}{K} J(t,k)(y(t,k) - y(t) + e(k))$$

$$= \sum_{t \in [T-1]} \gamma(\lambda) J(0,0)(I - A(\lambda))^t y$$

$$+ \sum_{t \in [T-1],k} \frac{\gamma(\lambda)}{K}(J(t,k) - J(0,0))(I - A(\lambda))^t y$$

$$- \sum_{t,k} \frac{\gamma(\lambda)}{K} J(t,k)(y(t,k) - y(t) + e(k))$$

$$= B_1 + B_2 + B_3$$

where

$$B_1 = \sum_{t \in [T-1]} \gamma(\lambda) J(0,0)(I - A(\lambda))^t y$$

$$B_2 = \sum_{t \in [T-1],k} \frac{\gamma(\lambda)}{K}(J(t,k) - J(0,0))(I - A(\lambda))^t y$$

$$B_3 = \sum_{t,k} \frac{\gamma(\lambda)}{K} J(t,k)(y(t,k) - y(t) + e(k))$$

Then by substituing in the following claims, we have

$$\|W(T) - W(0)\|_F$$

$$\le (y^T A(\lambda)^{-T} H^\infty A(\lambda)^{-1} y)^{1/2} + \mathcal{O}\Big(\frac{n\sigma}{s_{min}} \cdot \text{poly}(\log(m/\delta)) + \frac{n^4}{\sigma^{1/2}m^{1/4}} \cdot \text{poly}(\log(m/\delta))\Big)$$

$\square$

**Claim 1.** *With probability at least $1 - \delta$ over random initialization, as $t \to \infty$, we have*

$$\|B_1\|_2^2 \leq y^T (A(\lambda)^{-1})^T H^\infty A(\lambda)^{-1} y + \mathcal{O}(\frac{n^2 \sqrt{\log(n/\delta)}}{s_{min}^2 \sqrt{m}})$$

**Claim 2.** *With probability at least $1 - \delta$ over random initialization, we have*

$$\|B_2\|_2 \leq \frac{n^{3/2} poly(\log(m/\delta))}{m^{1/4} \sigma^{1/2} s_{min}^{3/2}}$$

**Claim 3.**

$$\|B_3\|_2 \leq (\frac{n\sigma}{s_{min}} + \frac{n^4}{s_{min}^3 \sigma \sqrt{m}}) \cdot poly(\log(m/\delta))$$

The above three claims are the slightly modified version of Claim C.8-10 in Huang et al. (2021), So the proofs from there naturally extend to the proofs of these three claims.

**Theorem 3.** *For $T$ sufficiently large, $\sigma = \mathcal{O}(\lambda poly(\log n, \log(1/\delta))/n))$, $m = \Omega(\sigma^{-2}(n^{16} poly(\log n, \log(1/\delta), \lambda^{-1})))$, the loss function $l$ being 1-Lipschitz in its first argument, then with probability at least $1 - \delta$ over the random initialization, the population loss $L_\mathcal{D}(f)$ of the global model $W := \frac{1}{N} \sum_c W_c$ is upper bounded by*

$$L_\mathcal{D}(f) \leq \sqrt{2y^T A(\lambda)^{-T} H^\infty A(\lambda)^{-1} y/n} + \mathcal{O}\big(\sqrt{\log(n/s_{min}\delta)/2n}\big)$$

*where $A(\lambda) = (1 - \lambda)\frac{\eta_l \eta_g K}{N} H^\infty + \lambda \eta_l K \tilde{H}_c^\infty$.*

*Proof.* This is an extension of the result in Huang et al. (2021), by substituting lemma 5 into the proof of Theorem C.11 in Huang et al. (2021). □

## C.4 CLIENT LEVEL

For further inspection on the algorithm, we consider a client level generalization. That is, we consider the generalization bound on a client's parameter $W_c$. Note that:

$$vec(W_c(t+1)) = vec(W_c(t)) - \lambda\eta_l \sum_k J_c(t,k)(y_c(t,k) - y_c) - (1-\lambda)\frac{\eta_l \eta_g}{N} \sum_k J(t,k)(y(t,k) - y)$$

We first define a matrix that will be used later.

**Definition 6** (Cross Gram matrix).

$$H_c^\times(t,k)_{i,j} = \frac{1}{m} \sum_r x_i^T x_j \mathbb{1}_{w_{0,r}^T x_i \geq 0, w_{c,r}^T(t,k)x_j \geq 0} \qquad \in \mathbb{R}^{n_c \times n}$$

*where $i \in S_c$ and $j$ spans all clients.*

This matrix describes the effect of the global update on the client's local model.

**Theorem 4.** *For $T$ sufficiently large, $\sigma = \mathcal{O}(\lambda poly(\log n, \log(1/\delta))/n))$, $m = \Omega(\sigma^{-2}(n^{16} poly(\log n, \log(1/\delta), \lambda^{-1})))$, the loss function $l$ being 1-Lipschitz in its first argument, then with probability at least $1 - \delta$ over the random initialization, the population loss $L_D(f)$ of the client model $W_c$ is upper bounded by*

$$L_\mathcal{D}(f) \leq \sqrt{2y^T A(\lambda)^{-T} G_c(\lambda) A(\lambda)^{-1} y/n} + \mathcal{O}\big(\sqrt{\log(n/s_{min}\delta)/2n}\big)$$

*where $G_c(\lambda) = (1 - \lambda)^2 \frac{\eta_l^2 \eta_g^2 K^2}{N^2} H + \lambda^2 \eta_l^2 K^2 H_c + \lambda(1 - \lambda)\eta_l^2 \eta_g K^2 H_c^\times$.*

This theorem is a natural extension of theorem 3, and the proof also naturally extends. Further discussion is left for future work.

## APPENDIX D    EXPERIMENTS

In this section, we show more experimental results and implementation details. Sec. D.1 gives more details for implementation, including dataset details, model architectures, training, and testing details. Sec. D.2 shows more experiment results, including the performance regarding various batch sizes, the complete evaluation metrics on the binary classification on the Camelyon17 dataset.

### D.1    EXPERIMENTAL DETAILS

We here introduce the complete details of datasets, data splitting, and implementation details.

**Digits5.** *Digits-5* zhou2020learning,fedbn with digits images showing drastic differences in font style, color, and background. We take each data source/style as a client. We train a 6-layer convolutional neural network for the digit classification, specifically, the model has 3 convolutional layers and 3 fully-connected layers, we further add batch normalization layers after the first five layers to fit the requirements of FedBN. We use the SGD optimizer with a learning rate of 0.01 and batch size of 128. The loss function is the Cross Entropy loss. The total number of training rounds is 100 with a local update epoch of 1. All input images are resized to $28 \times 28$.

**Office-Caltech10.** *Office-Caltech10* gong2012geodesic contains images acquired in different cameras or environments, with four different data sources in total. We take ResNet-18 as the backbone and use the SGD optimizer with a learning rate of 0.01 and batch size of 32. The loss function is the Cross Entropy loss. The total number of training rounds is 200 with a local update epoch of 1. The input images are normalized using the mean and std of Imagenet in PyTorch, which is specifically mean = [0.485, 0.456, 0.406] and std = [0.229, 0.224, 0.225]. All input images are resized to $256 \times 256$

**DomainNet.** *DomainNet* peng2019moment has images with different image styles (clipart, infograph, painting, quickdraw, real, and sketch). Following FedBN, we choose the top-10 class based on data amount from DomainNet containing images over 345 categories for simplicity. The training settings are the same as the Office-Caltech10 dataset, and we change the training round from 200 to 100, since the model converges faster on the DomainNet dataset.

**Camelyon17.** *Camelyon17* bandi2018detection shows histology images with different stains from 5 hospitals. All histopathology images are stained with the H.E. staining and show various appearances. We use the DenseNet121 as the backbone, SGD optimizer with a learning rate of 0.01, and batch size of 32. We train the model for 40 rounds in total, and the local update epoch is 1. The image input size is $96 \times 96$. The loss function is the Cross Entropy loss. Note that this dataset is a very large dataset which contains over 450,000 histology images.

**Retinal.** *Retinal* fundus dataset contains retinal fundus images acquired from 6 different institutions (Fumero et al., 2011; Sivaswamy et al., 2015; Almazroa et al., 2018; Orlando et al., 2020). We use the U-Net for segmentation, the optimizer is Adam with a learning rate of $1e^{-3}$ and $\beta = (0.9, 0.99)$. We train the model for 100 communication rounds in total with a local update epoch of 1. The batch size is 8. We use the dice loss and report both the Dice score and HD distance. All images are resized to $256 \times 256$.

For all datasets, we take each data source as one client and split the data of each client into train, validation, and testing sets with a ratio of 0.6, 0.2, and 0.2. We choose the best model based on the validation data and report the test performance accordingly. Code will be released after acceptance.

### D.2    MORE EXPERIMENTS

Here we present more experiment results, which mainly include two parts. The first part is the study on the effects of batch size on our method's performance, and the second part is the complete evaluation results on the Camelyon17 dataset.

**Effects of batch sizes.** As our implementation accumulates the feature matrix iteratively during local steps (Line 7 in the algorithm box), the batch size may slightly change the values of the feature matrix. In this case, we further explore the performance changes by using different batch sizes (4,8,16,32,64,128) on the Digit5 dataset. The results are shown in Table. 6. From the results it can be observed that changing batch size has very mild effects on the final performance, the overall accuracy

Table 6: Performance using different batch sizes on the Digits dataset.

| Batchsize | MNIST | SVHN | USPS | Synth | MNISTM | Average |
|---|---|---|---|---|---|---|
| 8 | 99.40 | 93.53 | 98.92 | 99.70 | 97.19 | 97.75 |
| 16 | 99.49 | 93.85 | 98.87 | 99.68 | 97.41 | 97.86 |
| 32 | 99.40 | 93.37 | 99.03 | 99.62 | 96.86 | 97.66 |
| 64 | 99.37 | 92.87 | 98.87 | 99.50 | 96.14 | 97.35 |
| 128 | 99.25 | 92.17 | 98.71 | 99.42 | 95.71 | 97.05 |

changes are less than $1\%$. This further supports our implementation of iteratively accumulating the feature matrix during local SGD, which takes less computational cost than re-calculate all samples again after 1 local epoch.

**Complete evaluation on Camelyon17.** As the classification task on the Camelyon17 dataset is a binary classification, so we further report the full evaluation metrics, including the Accuracy, AUC, sensitivity, specificity, and the F1-score. From the Table. 7, it can be observed that our proposed method consistently outperforms all compared methods regarding all metrics.

Table 7: Complete evaluation metrics on the Camelyon17 dataset.

| | Accuracy | AUC | Sensitivity | Specificity | F1-score |
|---|---|---|---|---|---|
| FedAvg | 95.30 | 98.90 | 94.91 | 95.69 | 95.30 |
| APFL | 96.42 | 99.35 | 94.42 | 98.42 | 96.42 |
| L2SGD | 95.93 | 99.22 | 95.12 | 96.73 | 95.92 |
| FedAlt | 97.91 | 99.57 | 96.55 | 97.90 | 97.22 |
| PerFedAvg | 96.08 | 99.24 | 95.37 | 96.79 | 96.08 |
| FedBN | 95.77 | 99.15 | 95.02 | 96.53 | 95.77 |
| FedFOMO | 95.62 | 99.15 | 95.35 | 95.90 | 95.62 |
| FedRep | 96.78 | 99.39 | 96.78 | 96.79 | 96.78 |
| FedBABU | 95.77 | 99.15 | 95.03 | 96.52 | 95.77 |
| FedHKD | 95.89 | 98.43 | 93.89 | 94.67 | 94.28 |
| LG-Mix (Ours) | **98.75** | **99.89** | **98.63** | **98.87** | **98.75** |

**Loss and accuracy curves.** We have further analyzed the convergence speed of our method by comparing the training loss, validation loss and validation accuracy on the Digits5 dataset. We select the top 3 ranked methods on the Digits5 dataset, and the results are shown in Fig. 4. From the figure, it can be observed that other methods show quicker convergence speed and higher validation performance than the baseline method FedAvg, and our method further promotes the convergence speed, especially the training loss. These curves further demonstrate the efficacy on the convergence rate of our proposed method.

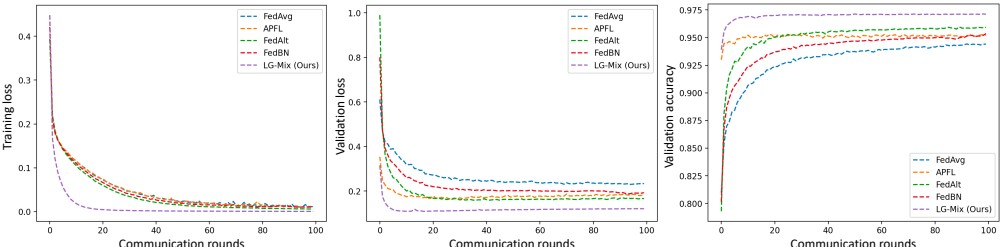

Figure 4: Training loss, validation loss, and validation accuracy. The comparison includes the FedAvg, the top 3 ranked SOTA methods, and our method on the Digits5 dataset.

**Training time costs.** We further investigate the training time costs and report the wall-clock time by comparing our method and others on all tasks. The GPU we used for training is GeForce RTX

Table 8: Wall-clock time (hours) for different methods on different tasks.

| Method | Digits5 | OfficeCaltech10 | DomainNet | Camelyon17 | Retinal |
|---|---|---|---|---|---|
| FedAvg | 3.70 | 1.28 | 17.37 | 55.01 | 25.95 |
| APFL | 4.38 | 1.70 | 19.59 | 79.12 | 26.36 |
| L2SGD | 3.83 | 1.40 | 18.42 | 57.17 | 26.79 |
| FedAlt | 4.09 | 1.47 | 28.77 | 76.80 | 34.16 |
| PerFedAvg | 3.79 | 1.45 | 18.66 | 56.97 | 26.44 |
| FedBN | 3.73 | 1.26 | 17.43 | 55.03 | 26.56 |
| FedFOMO | 6.02 | 2.80 | 19.04 | 76.80 | 36.90 |
| FedRep | 4.32 | 1.41 | 19.18 | 55.77 | - |
| FedBABU | 3.96 | 1.31 | 15.92 | 62.12 | - |
| FedHKD | 22.43 | 2.64 | 37.33 | 68.71 | - |
| LG-Mix (Ours) | 5.86 | 1.32 | 18.92 | 57.98 | 26.55 |

Table 9: GPU memory cost (MB) comparison on different tasks.

| Method | Digits5 | OfficeCaltech10 | DomainNet | Camelyon17 | Retinal |
|---|---|---|---|---|---|
| FedAvg | 270.24 | 1042.32 | 1042.32 | 1096.89 | 1814.29 |
| APFL | 600.77 | 2993.48 | 2993.48 | 2955.24 | 4504.02 |
| L2SGD | 456.18 | 2057.62 | 1157.62 | 2072.09 | 2329.91 |
| FedAlt | 458.27 | 1042.33 | 1042.33 | 1096.89 | 2297.57 |
| PerFedAvg | 270.24 | 1042.32 | 1042.32 | 1096.89 | 1814.29 |
| FedBN | 458.27 | 1042.32 | 1042.32 | 1097.99 | 2295.21 |
| FedFOMO | 458.28 | 1948.98 | 1948.98 | 1985.06 | 2837.97 |
| FedRep | 458.27 | 1042.30 | 1042.30 | 1096.89 | - |
| FedBABU | 458.27 | 1042.30 | 1042.30 | 1096.89 | - |
| FedHKD | 864.28 | 3589.56 | 3589.56 | 3021.02 | - |
| LG-Mix (Ours) | 472.37 | 2101.01 | 2101.01 | 1905.38 | 3730.69 |

2080 Ti. The results are shown in Table 8. We report the training ours. From the table, we can see FedHKD takes a significant time cost than other methods, which is because it requires performing an extra validation process using different client models, in order to generate the hyper-knowledge. For method APFL, it needs to take an extra forward pass and also optimize the mixing ratio factor in their method by calculating the parameter differences. Please note that we do not report the time cost of FedRep, FedBABU, and FedHKD on the retinal dataset, because they are specifically designed for the classification task. Compared with all these methods, our proposed method shows a reasonable computational time cost. The cost is slightly higher than some baseline methods, but it is a trade-off between the computational cost and performance.

**GPU memory cost.** We have investigated the GPU memory cost by comparing our method and others. Compared with FedAvg, our method additionally stores a copy of the global model during local training, and it calculates the trace of latent feature matrices. We have tracked the peak GPU memory cost of each method and list the costs on different tasks in Table 9. It can be observed that APFL and FedHKD show significantly higher GPU memory costs than others, which is because they require more memory to store the local copy of the global model. Specifically, APFL stores the local model, the local copy of the global model, and a local personalized model. FedHKD requires storing local models from other clients to generate the hyper knowledge for distillation. Our method lies in a reasonable range of GPU memory cost while presenting higher performance. Please note that we do not report FedRep, FedBABU, and FedHKD on the retinal dataset because they are specifically designed for the classification task.

**Extended experiments on the BCI dataset.** We extend our evaluation from the image domain to the Brain-Computer Interface (BCI) data, which consists of classifying the mental imagery EEG datasets. Specifically, we use four datasets from the MOABB benchmark (Jayaram & Barachant, 2018). We choose four datasets (AlexMI, BNCI2014_001, BNCI2015_004, Zhou2016) which contain common classes of right hand and feet. We estimated the covariance matrix representations

Table 10: Comparison on the BCI dataset.

| Method | Accuracy | AUC | Sensitivity | Specificity | F1-score |
|---|---|---|---|---|---|
| FedAvg | 67.38 | 69.71 | 68.14 | 66.77 | 67.29 |
| APFL | 71.01 | 75.48 | 71.21 | 71.08 | 70.97 |
| L2SGD | 67.39 | 69.96 | 69.11 | 65.88 | 67.31 |
| FedAlt | 68.46 | 72.74 | 69.72 | 67.52 | 68.45 |
| PerFedAvg | 67.47 | 71.25 | 64.22 | 70.56 | 67.35 |
| FedBN | 68.06 | 70.62 | 68.28 | 68.03 | 68.03 |
| FedFOMO | 64.69 | 68.56 | 65.36 | 64.18 | 64.53 |
| FedRep | 68.10 | 69.82 | 69.75 | 66.68 | 68.01 |
| FedBABU | 68.03 | 70.45 | 69.99 | 66.25 | 67.95 |
| FedHKD | 68.69 | 70.34 | 64.99 | 72.32 | 68.44 |
| LG-Mix (Ours) | **73.74** | **78.11** | **75.19** | **72.58** | **73.72** |

of each EEG signal as a feature and performed the tangent space projection for the matrices. The final input of each dataset is a 1-d vector, and we further add zero-paddings to have the same input dimension. We compared our method with others and reported the performance under five metrics in Table 10. From the table, we can observe that PFL methods show better performance than the FedAvg on such heterogeneous features, while our method outperforms all compared methods on five metrics. This further validates the efficacy of our method on 1-d signal data.

## APPENDIX E  DISCUSSIONS

In this paper, we present a novel PFL approach aimed at addressing the challenges posed by heterogeneous features during the training process. Our proposed method incorporates considerations of the convergence rate of both local and global models. To achieve the model personalization, we take the trace of the Gram matrix using local/global updates for gradient descent as an approximation of the convergence rate. By employing the ratio of the trace, we perform a linear combination of local and global updates. Additionally, due to the substantial increase in computational costs associated with the Gram matrix as the number of samples grows, our implementation further approximates the calculation by calculating the trace of a latent feature matrix. The latent feature is extracted from the last hidden layer. However, our method only considers the feature matrix to mix local and global updates. While this approximation is reasonable and reflects the convergence rate in principle, it may not be an accurate estimation when using large foundation models as the model backbone. When using large foundation models, such as pre-trained deep neural networks, the convergence behavior may differ due to the model's architecture and complexity. In such cases, relying solely on the last-layer latent feature for trace approximation might not capture the true convergence rate accurately. Therefore, there is a risk of obtaining less precise estimations in these scenarios. To address this limitation and obtain more comprehensive measurements for combining local and global updates, one potential approach is to consider the prediction errors of local and global models. By incorporating the model errors into the mixing process, we can potentially gain insights into the convergence behavior of the models. This approach would provide a more holistic perspective on the convergence rate and guide the combination of local and global updates more effectively. However, exploring the use of large foundation models in federated learning is still an open question. The unique challenges and considerations introduced by these models need to be thoroughly investigated. Future research should focus on validating the effectiveness of our method when applied to federated learning scenarios involving large foundation models.

Our method aims to improve the personalization of client models for heterogeneous features. This can have benefits in various applications, such as healthcare, where medical images collected from different hospitals are heterogeneous. By improving the model personalization, our method can potentially improve the accuracy and effectiveness for better healthcare outcomes. Our work also has several potential implications for future research. First, our approach provides a new perspective on addressing heterogeneous features in PFL, and can be extended to segmentation asks, whereas many existing PFL methods are only validated on classification tasks. Second, our work proposes

an NTK viewpoint for analyzing our PFL method by combining local and global updates, it will be interesting to extend the NTK framework for analyzing the general PFL frameworks.

