# OpenReview forum: "Heterogeneous Personalized Federated Learning by Local-Global Updates Mixing via Convergence Rate"
_ICLR.cc/2024/Conference — ICLR 2024 poster_

### Official Review · Reviewer_Ho8d · 2023-10-17

**Soundness:** 2 fair
**Presentation:** 2 fair
**Contribution:** 2 fair
**Rating:** 5
**Confidence:** 2

**Summary:**

The paper addressed the issue of improving convergence of personalized federated learning using a mix of local and global model updates. The optimal mix is found through a measure related to the neural tangent kernel / trace of the Gram matrix. An approximation of the trace is introduced based on taking only the last layer into account. The method is demonstrated on several image problems.

**Strengths:**

The idea of mixing local and global model updates in a manner that optimizes the convergence in PFL is simple and intuitive. Making use of the Gram trace in this context makes good sense.

The proposed method is compared with many other relevant methods.

**Weaknesses:**

The problem setup could be more clearly defined from the beginning. The paper is motivated by a discussion of the challenges of data heterogeneity; however, it would be more clear if the authors in the beginning of the paper would include a clear definition of the type of heterogeneity that is addressed, and which personalized settings the developed methods apply to. Statements such as "To solve the data heterogeneity problem..." seem to indicate that heterogenous data is a kind of external obstacle, rather than an inherent property of the problem PFL is aiming to solve.

In the related works section, there is the following statement: "However, the existing approaches have primarily focused on addressing heterogeneity related to class distributions, system resources, and model architecture. The issue of feature distributional shifts in PFL remains under-explored." While it may be true that the issue is underexplored, it would be great with a clear delineage of which existing works address this issue, and which of the related methods could reasonably expected to work under covariate shift.

"In FL, each client will update the local model using the global update after each communication
round." While this is not wrong per se, there are many ways local and global updates are carried out in different algorithms, depending on the problem setup and assumptions behind the particular algorithm.

It is not clear if the results are a fair comparison, i.e. whether or not each algorithm is independently optimized sufficiently well.

The section on the convergence analysis seems a bit disintegrated with the rest of the paper.

Simple experiments that highlight the applicability of the method are not included. I would have appreciated a clear demonstration, e.g. in comparison with fixed local-global ratios.

An empirical demonstration of proposition 1 would have been a strong argument for the specific choices made for the mixing ratio.

Minor issues
paramters -> parameters
footnote -> subscript
tr should be roman in eq. 7

**Questions:**

Is this critique fair: "However, despite advancements, these methods aim to learn a common
global model, but it is challenging to ensure the consensus global model is best-for-all."? It is my understanding that the objective of these methods is to learn a global model, so they are designed for another purpose than PFL.

Are the methods described only applicable in the context of stochastic gradient descent? It would be beneficial to have a clear definition of the problem setup in which the proposed methods apply.

In sec. 3.1 both covariate shift and concept drift are mentioned. Is this method applicable to both scenarios? This should preferably be clear from the introduction.

Are there any previous papers that consider the update strategy in eq. 3, or is this an idea that has not yet been explored?

It is not always clear to me if w_c and u denote models or model weights?

Where does eq. 4 come from? There is no details regarding its derivation or any clear reference.

Is proposition 1 a novel result, and if so in which sense?

It is uncler to me if the mixing ratio in eq. 5 is optimal in some sense? Intuitively, the global and local model might both have different error and different convergence properties, so should the model error not also be taken into account?

Does it make a big difference that you only consider the last layer in the approximation of the Gram matrix? Can the arguments be supported empirically or theoretically?

What exactly is meant by this statement: "since the data is distributed and not directly accessible, we propose employing features from local data in conjunction with the global model u"? Do you employ the representation (first layers) of the local model?

Is it correctly understood that you use a fixed learning rate for all experiments, which is not optimized?

What are the practical implications of the convergence analysis in sec. 4? How is the theorem and proof different form exising convergence results?

I do not understand the arguments in the paragraph titled "Feature value distribution by our personalized model". What results is the statement "If a model learns confident representations, then the related neurons should be highly activated..." based on?

If instead we used a fixed (optimized) ratio of local and global updates, does your method have a clear advantage?

---

> ### Author Response · Authors · 2023-11-21
> **Rebuttal by authors (1/3)**
>
> We sincerely thank you for your positive comments on our idea, method that makes good sense, and the experiments where our method is compared with many other relevant methods. Thank you for providing us with numerous constructive comments and suggestions. After carefully reviewing your comments, we found your concerns regarding the writing can be addressed via minor changes. We uploaded a revised version and highlighted the changes in blue color.
> In the following, we would like to address your questions regarding further clarification of our method and analysis.
>
> > - It is not clear if the results are a fair comparison, i.e. whether or not each algorithm is independently optimized sufficiently well.
>
> Our comparative analysis is performed in a fair manner, as all the methods follow the same experimental settings. Furthermore, the hyperparameters of the compared method are carefully fine-tuned. The reported findings presented in this study are in alignment with recent publications [1, 2], further substantiating the credibility and reliability of our results.
>
> [1] Huang, Wenke, Mang Ye, and Bo Du. "Learn from others and be yourself in heterogeneous federated learning." In Proceedings of the IEEE/CVF Conference on Computer Vision and Pattern Recognition, pp. 10143-10153. 2022.
>
> [2] Zhuang, Weiming, and Lingjuan Lyu. "Is Normalization Indispensable for Multi-domain Federated Learning?." arXiv preprint arXiv:2306.05879 (2023).
>
> > - The section on the convergence analysis seems a bit disintegrated with the rest of the paper.
>
> We propose Theorem 1 in the convergence analysis to give a bound, while our method is motivated by Proposition 1. Theorem 1 serves as a more general proof of the convergence of the PFL algorithms combining local and global updates, as it poses no constraint on $\lambda$. We admit that more discussion can be made on the convergence analysis, and we have added it to the manuscript.
>
> > - Simple experiments that highlight the applicability of the method are not included. I would have appreciated a clear demonstration, e.g. in comparison with fixed local-global ratios.
>
> Following your suggestion, we have extended the investigation on the performance by using different fixed ratios and the optimized ratio. For the fixed optimized ratio, we take the average ratio of our method in the last 10 training rounds. The results are shown in the following table. When the ratio equals to 0, our method will degenerate to the FedAvg, and when the ratio goes to 1, it equals to complete client local training. From the results, it can be observed that introducing the mixing of local and global updates helps improve the personalized model performance, and our dynamic adjustment shows the highest accuracy.
> | Mixing ratio |   0   |  0.25 |  0.5  |  0.75 |  1.0  | Fix(our optimized) | Ours  |
> |:------------:|:-----:|:-----:|:-----:|:-----:|:-----:|:-----:|-------|
> |   Accuracy   | 94.68 | 95.37 | 95.64 | 94.48 | 96.03 |  96.65 | 97.11|
>
> >- An empirical demonstration of proposition 1 would have been a strong argument for the specific choices made for the mixing ratio.
>
> Thank you for your insightful comment. Our proposition 1 mainly motivates us to approximate the convergence rate by measuring the trace of the Gram matrix. However, the computation of the Gram matrix $H$ in practical networks poses significant challenges and is often computationally infeasible [1,2,3].
>
> [1] Fort, Stanislav, Gintare Karolina Dziugaite, Mansheej Paul, Sepideh Kharaghani, Daniel M. Roy, and Surya Ganguli. "Deep learning versus kernel learning: an empirical study of loss landscape geometry and the time evolution of the neural tangent kernel." Advances in Neural Information Processing Systems 33 (2020): 5850-5861.
>
> [2] Mohamadi, Mohamad Amin, Wonho Bae, and Danica J. Sutherland. "A fast, well-founded approximation to the empirical neural tangent kernel." In International Conference on Machine Learning, pp. 25061-25081. PMLR, 2023.
>
> [3] Holzmüller, David, Viktor Zaverkin, Johannes Kästner, and Ingo Steinwart. "A framework and benchmark for deep batch active learning for regression." Journal of Machine Learning Research 24, no. 164 (2023): 1-81.

---

> ### Author Response · Authors · 2023-11-21
> **Rebuttal by authors (2/3)**
>
> > - Are the methods described only applicable in the context of stochastic gradient descent? It would be beneficial to have a clear definition of the problem setup in which the proposed methods apply.
>
> Yes, our analysis is based on the NTK, which is only applicable in the context of gradient descent.
>
> > - In sec. 3.1 both covariate shift and concept drift are mentioned. Is this method applicable to both scenarios? This should preferably be clear from the introduction.
>
> Yes, our method is applicable to both covariate shift and concept drift. We have clarified this in Section 3.1. Our experimental evaluation also includes both covariate shift and concept drift.
>
> > - Are there any previous papers that consider the update strategy in eq. 3, or is this an idea that has not yet been explored?
>
> The combination of local and global updates in PFL is not an uncommon practice, we have discussed the one related work APFL in the fourth paragraph in the introduction. However, our aim is to find a smart way to dynamically adjust the combination weight. To determine the weight, we draw inspiration from NTK. Our idea is to take the trace of the Gram matrix as a hint to determine the mixing ratio, which has not been explored yet.
>
> > - It is not always clear to me if w_c and u denote models or model weights?
>
> We acknowledge that the notation we initially employed needed to be clarified. We have rectified this issue in the updated manuscript, by using $w$ and $u$ to denote the model weights, and $y(t)=f(w(t),x)$ denotes the model prediction with weights $w(t)$ and $f: \mathbb{R}^d \rightarrow \mathbb{R}$ is the model function.
>
> >- Where does eq. 4 come from? There is no details regarding its derivation or any clear reference.
>
> Eq. 4 comes from Du et al.’s work [1], which uses the Gram matrix to characterize the optimization process of gradient descent for neural networks. This result is originally presented in Section 4 of "Gradient Descent Finds Global Minima of Deep Neural Networks" in the paper.
>
> [1] Du, Simon S., Xiyu Zhai, Barnabas Poczos, and Aarti Singh. "Gradient descent provably optimizes over-parameterized neural networks." ICLR (2019).
>
> > - Is proposition 1 a novel result, and if so in which sense?
>
> Regarding the novelty, proposition 1 provides new insight to motivate our idea, the intuition behind this proposition is different from conventional convergence theory. Our algorithm has demonstrated the effectiveness of using trace as an approximation of the convergence rate in practice. However, the accuracy of this approximation requires further investigation, which can be left for future work.
>
> > - It is uncler to me if the mixing ratio in eq. 5 is optimal in some sense? Intuitively, the global and local model might both have different error and different convergence properties, so should the model error not also be taken into account?
>
> It would be inaccurate to claim that the ratio presented in Eq. 5 is optimal. Instead, it is a ratio derived from theoretical intuition and has been empirically verified to yield significant improvements.
>
> We agree with you that model error between the local and global models can also affect the convergence property. Further research can be done to investigate whether taking the prediction error into account could speed up the convergence even further.
>
> > - Does it make a big difference that you only consider the last layer in the approximation of the Gram matrix? Can the arguments be supported empirically or theoretically?
>
> The approximation should make no big differences in principle. Our approximation is supported by Seleznova et al.’s work [1], where they demonstrate that the empirical NTK with entries of last-layer features highly aligns with the original NTK regarding training dynamics. Their findings are supported both theoretically and empirically.
>
> [1] Seleznova, Mariia, Dana Weitzner, Raja Giryes, Gitta Kutyniok, and Hung-Hsu Chou. "Neural (Tangent Kernel) Collapse." arXiv preprint arXiv:2305.16427 (2023).
>
> > - What exactly is meant by this statement: "since the data is distributed and not directly accessible, we propose employing features from local data in conjunction with the global model u"? Do you employ the representation (first layers) of the local model?
>
> To calculate the trace, we employ the representation of both local and global models. Specifically, each client employs its local model to calculate the local feature, while also maintaining a copy of the global model to compute the global feature. The feature representation is extracted from the last hidden layer, that is, parameters up to the penultimate layer.

---

> ### Author Response · Authors · 2023-11-21
> **Rebuttal by authors (3/3)**
>
> > - Is it correctly understood that you use a fixed learning rate for all experiments, which is not optimized?
>
> For classification tasks, we follow the FedBN [1] to set the learning rate as $1e-2$. For the segmentation task, we use the learning rate of $1e-3$. We did a grid search to determine an appropriate range for the learning rate.
>
> [1] Li, Xiaoxiao, Meirui Jiang, Xiaofei Zhang, Michael Kamp, and Qi Dou. "Fedbn: Federated learning on non-iid features via local batch normalization." International Conference on Learning Representations (2021).
>
> >- What are the practical implications of the convergence analysis in sec. 4? How is the theorem and proof different from existing convergence results?
>
> The main point is it governs the convergence of our algorithm. The proof adapted the same method used in FL-NTK [1], whereas FL-NTK only shows the convergence of vanilla FedAvg, and we further extended the result to show the convergence for PFL algorithms with combined local and global updates. To be more specific, Prop.2 and Prop.3 in Appendix B.5 Proof of Theorem 1 are directly adapted from FL-NTK, which illustrates the effect of global and local updates on global and local models respectively. Our work mainly focuses on bounding the fluctuation caused by the cross effect, as shown in Prop.4 and Prop.5.
> We have added more discussion on our convergence analysis in the updated version.
>
> [1] Huang, Baihe, Xiaoxiao Li, Zhao Song, and Xin Yang. "Fl-ntk: A neural tangent kernel-based framework for federated learning analysis." In International Conference on Machine Learning, pp. 4423-4434. PMLR, 2021.
>
> > - I do not understand the arguments in the paragraph titled "Feature value distribution by our personalized model". What results is the statement "If a model learns confident representations, then the related neurons should be highly activated..." based on?
>
> Our statement is supported by Bau and Zhou et al.'s studies [1, 2]. They have demonstrated the strong correlation between neuron activation values and representation learning. Specifically, their studies show that neurons that effectively learn important representations exhibit high activation values. Consequently, forcibly deactivating these neurons results in a substantial drop in performance. We have included this reference in our manuscript.
>
> [1] Bau, David, Jun-Yan Zhu, Hendrik Strobelt, Agata Lapedriza, Bolei Zhou, and Antonio Torralba. "Understanding the role of individual units in a deep neural network." Proceedings of the National Academy of Sciences 117, no. 48 (2020): 30071-30078.
>
> [2] ​​Zhou, Bolei, David Bau, Aude Oliva, and Antonio Torralba. "Interpreting deep visual representations via network dissection." IEEE transactions on pattern analysis and machine intelligence 41, no. 9 (2018): 2131-2145.

---

### Official Review · Reviewer_efqn · 2023-10-22

**Soundness:** 3 good
**Presentation:** 3 good
**Contribution:** 2 fair
**Rating:** 6
**Confidence:** 3

**Summary:**

This paper proposes a mixing local initialization in pFL studies via the ratio of traces of the local feature matrix and global feature matrix. The proposed method experimentally and efficiently improves the final performance. Ablation studies are also introduced to learn the efficiency of the mixing rate.

**Strengths:**

1. Writing is clear and easy to follow.
2. The motivation of local initialization is sound and efficient in the pFL community. I appreciate the author's comparative analysis of ablation studies to verify the details of the proposed algorithm in the training process.
3. A brief convergence analysis is provided.

**Weaknesses:**

1. In the Algorithm Line.6, how to calculate the $h(x_i)$? This feature seems to adopt the global feature extractor to generate the approximation of global convergence. While in the local client, this seems to be untenable without the access to the global model. Does it require storing a global model locally? If so, I think a storage comparison among baselines is required to comprehensively compare the efficiency.
2. The authors should provide the training wall-clock time required to compare the practical efficiency.
3. The motivation is to select the faster convergence to match the local or global model. So I think it is necessary to introduce the loss curve or gradient norm to illustrate its improvements. This paper does not analyze the improvements in the test error and generalization performance. Under the specific convergence analysis, how does the proposed method match the convergence analysis? And does the training loss or gradient norm decrease faster?

**Questions:**

Thanks for the great submission but I still find there are some important issues to be resolved in the current version. My concerns are lined in the weaknesses.

---

> ### Author Response · Authors · 2023-11-21
> **Rebuttal by authors**
>
> We sincerely thank you for your positive comments on our paper writing, method motivation, and the comparative analysis of ablation studies to verify the details of our method. We would like to address each of your questions below.
>
> >- How to calculate the h(x_i)? Does it require storing a global model locally? And adding a storage comparison among baselines.
>
> **(a) Calculate the $h(x_i)$:**
> The $h(x_i)$ is the latent feature representation of the input $x_i$ extracted from the last hidden layer, where $h$ denotes the parameters up to the penultimate layer.
>
>
> **(b) Store a global model locally:**
> Yes. A global model is required to keep locally to calculate the global feature.
>
> **(c) Storage comparison among baselines：**
> Following your suggestion, we monitored the peak GPU memory cost of each method and listed the costs in the table below. Our method takes a reasonable cost compared with other baseline methods. Please note that we do not report FedRep, FedBABU, and FedHKD on the retinal dataset because they are specifically designed for classification tasks.
>
> **GPU memory cost (MB)**
> |     Method    | Digits5 | OfficeCaltech10 | DomainNet | Camelyon17 | Retinal |
> |:-------------:|:-------:|:---------------:|:---------:|:----------:|:-------:|
> |     FedAvg    |  270.24 |     1042.32     |  1042.32  |   1096.89  | 1814.29 |
> |     APFL      |  600.77 |     2993.48     |  2993.48  |   2955.24  | 4504.02 |
> |     L2SGD     |  456.18 |     2057.62     |  1157.62  |   2072.09  | 2329.91 |
> |     FedAlt    |  458.27 |     1042.33     |  1042.33  |   1096.89  | 2297.57 |
> |   PerFedAvg   |  270.24 |     1042.32     |  1042.32  |   1096.89  | 1814.29 |
> |     FedBN     |  458.27 |     1042.32     |  1042.32  |   1097.99  | 2295.21 |
> |    FedFOMO    |  458.28 |     1948.98     |  1948.98  |   1985.06  | 2837.97 |
> |     FedRep    |  458.27 |     1042.30     |  1042.30  |   1096.89  |    -    |
> |    FedBABU    |  458.27 |     1042.30     |  1042.30  |   1096.89  |    -    |
> |     FedHKD    |  864.28 |     3589.56     |  3589.56  |   3021.02  |    -    |
> | LG-Mix (Ours) |  472.37 |     2101.01     |  2101.01  |   1905.38  | 3730.69 |
>
> >- The training wall-clock time
>
> For the computational time, we counted the training time of all methods. The GPU type used for training is GeForce RTX 2080 Ti. From the results in the table, it can be observed that our method lies in a reasonable range with a slightly higher cost than FedAvg, which is a trade-off between computational efficiency and performance.
>
> **Training time (hour)**
> |     Method    | Digits5 | OfficeCaltech10 | DomainNet | Camelyon17 | Retinal |
> |:-------------:|:-------:|:---------------:|:---------:|:----------:|:-------:|
> |     FedAvg    |   3.70  |       1.28      |   17.37   |    55.01   |  25.95  |
> |     APFL      |   4.38  |       1.70      |   19.59   |    79.12   |  26.36  |
> |     L2SGD     |   3.83  |       1.40      |   18.42   |    57.17   |  26.79  |
> |     FedAlt    |   4.09  |       1.47      |   28.77   |    76.80   |  34.16  |
> |   PerFedAvg   |   3.79  |       1.45      |   18.66   |    56.97   |  26.44  |
> |     FedBN     |   3.73  |       1.26      |   17.43   |    55.03   |  26.56  |
> |    FedFOMO    |   6.02  |       2.80      |   19.04   |    76.80   |  36.90  |
> |     FedRep    |   4.32  |       1.41      |   19.18   |    55.77   |    -    |
> |    FedBABU    |   3.96  |       1.31      |   15.92   |    62.12   |    -    |
> |     FedHKD    |  22.43  |       2.64      |   37.33   |    68.71   |    -    |
> | LG-Mix (Ours) |   5.86  |       1.32      |   18.92   |    57.98   |  26.55  |
> ​​
>
>
> > - Draw the loss curve to illustrate the model improvements regarding faster convergence
>
> As per your suggestion, we have included the training and validation loss values in the table provided below. The results indicate that PFL methods generally converge faster than the baseline method FedAvg. Among all the compared methods, our method clearly promotes the convergence speed and presents the lowest training and validation loss. Additionally, we have draw the training/validation loss curves in Section D.2, Figure 4, in the Appendix for your reference.
>
>
> |  Round |    1   |    5   |   10   |   20   |   50   | 100    |
> |:------:|:------:|:------:|:------:|:------:|:------:|--------|
> | Training Loss|
> | FedAvg | 0.2062 | 0.1451 | 0.1145 | 0.0728 | 0.0239 | 0.0116 |
> |  APFL  | 0.2062 | 0.1453 | 0.1152 | 0.0753 | 0.0218 | 0.0107 |
> | FedAlt | 0.2208 | 0.1411 | 0.1049 | 0.0598 | 0.0146 | 0.0061 |
> | FedBN  | 0.2135 | 0.1431 | 0.1097 | 0.0663 | 0.0192 | 0.0111 |
> |   LG-Mix (Ours)  | 0.1682 | 0.0556 | 0.0183 | 0.0048 | 0.0015 | 0.0006 |
> | Validation Loss|
> | FedAvg | 0.4684 | 0.3906 | 0.3185 | 0.2716 | 0.2412 | 0.2335 |
> |  APFL  | 0.2457 | 0.2057 | 0.1739 | 0.1779 | 0.1772 | 0.1814 |
> | FedAlt | 0.4755 | 0.2596 | 0.2003 | 0.1676 | 0.1606 | 0.1655 |
> | FedBN  | 0.472  | 0.3251 | 0.2594 | 0.2196 | 0.2009 | 0.1922 |
> | LG-Mix (Ours)  | 0.1746 | 0.1162 | 0.1092 | 0.1103 | 0.1158 | 0.1203 |

---

> > ### Comment · Reviewer_efqn · 2023-11-22
> > **Thanks for the rebuttal**
> >
> > Thank you for the rebuttal. My main concerns have been checked in the additional experiments. I noticed that in the comparison of wall-clock time, the ratio of time consumption on different datasets seems to vary greatly. Why does this happen? For instance, FedFOMO on Camelyon17 requires 76.8h (1.39x to FedAvg). However, on DomainNet, it requires 19.04h only (1.09x to FedAvg).

---

> > > ### Author Response · Authors · 2023-11-22
> > > **Thank you for the reply**
> > >
> > > The time variation may be attributed to input/output (I/O) and CPU operations. The total duration of training, as measured by the wall-clock time, encompasses not only GPU training but also other activities involving I/O and CPU operations (e.g., disk I/O to read data, data preprocessing on CPU, and data transfer from CPU to GPU). While we utilize GPUs of the same type, it is crucial to note that they are installed on different computing servers, each equipped with different CPUs and other hardware components. Furthermore, the training time can be influenced by the dynamic workload experienced by these servers, introducing further variations.

---

> > > > ### Comment · Reviewer_efqn · 2023-11-22
> > > >
> > > > "While we utilize GPUs of the same type, it is crucial to note that they are installed on different computing servers, each equipped with different CPUs and other hardware components."
> > > >
> > > > Then this time consumption test should be very inaccurate. I think you should better update this experiment on a fair condition, e.g. for the same setups of the hardware. From the overall effect, the loss does not seem to be obvious. I have improved my score to 6 and I encourage the authors to finish the wall-clock time test in the revision.

---

> > > > > ### Author Response · Authors · 2023-11-22
> > > > > **Thank you for the reply**
> > > > >
> > > > > Many thanks for your suggestion and kind consideration. We will update this experiment by using the same computing server to avoid variation in other conditions.

---

### Official Review · Reviewer_i8zc · 2023-10-22

**Soundness:** 3 good
**Presentation:** 3 good
**Contribution:** 3 good
**Rating:** 8
**Confidence:** 2

**Summary:**

This paper proposes a novel approach to address the challenge of data heterogeneity in personalized federated learning. The proposed method leverages the convergence rate induced by Neural Tangent Kernel to quantify the importance of local and global updates, and subsequently mix these updates based on their importance. The authors have theoretically analyzed and experimentally evaluated their method on five datasets with heterogeneous data features in natural and medical images. The results show that the proposed method outperforms existing methods in terms of convergence rate and accuracy. Overall, this paper contributes to the development of more effective and efficient personalized federated learning methods.

**Strengths:**

The proposed Local-Global updates Mixing approach is original and innovative, leveraging the convergence rate induced by Neural Tangent Kernel to address the challenge of data heterogeneity in personalized federated learning. This approach is a creative combination of existing ideas and provides a new perspective on how to handle feature heterogeneity.

The clarity of the paper is excellent, with a clear and concise writing style, well-organized sections, and informative figures and tables.

**Weaknesses:**

the authors could discuss the scalability and robustness of their method to larger and more complex datasets, and explore potential applications in other domains beyond natural and medical images. The paper could benefit from a more detailed discussion of the limitations and potential extensions of the proposed method.

**Questions:**

1. how is eq. 4 derived? please give more explanantion on it. y is label, w(t) is the local model weights, what's the mearning of y-w(t)?
2. What's the cost (e.g., GPU memory cost, computational time) of the proposed method, compared to the baseline methods?
3. The proposed method is intuitive and interesting. What are the possible limitations of the proposed method, and how can they be addressed in future work?

**Details Of Ethics Concerns:**

null

---

> ### Author Response · Authors · 2023-11-21
> **Rebuttal by authors (1/3)**
>
> We sincerely thank you for your supportive comments on our approach novelty, paper clarity and organization, and informative results. Our detailed responses to your comments are as follows.
>
>
> > - The authors could discuss the scalability and robustness of their method to larger and more complex datasets, and explore potential applications in other domains beyond natural and medical images.
>
> We address your comments from three aspects: **(a) Clarify our existing large and complex datasets**, **(b) Extend the client scalability study (incorporating more clients) on a large complex medical dataset**, and **(c) Explore applications in the new domain of brain-computer interface data.**
>
>
> **(a)Clarify our existing large and complex datasets:**
> We included two real-world complex medical datasets in our experiments. The Camelyon17 dataset is a very large dataset that contains 450,000 histology image patches with different stains from five hospitals. The retinal dataset consists of retinal fundus images acquired from 6 different institutions with various imaging devices and protocols. The superior performance of our proposed method has demonstrated the robustness on real-world large and complex datasets.
>
>
> **(b)Extend the client scalability study (incorporating more clients) on a large complex medical dataset:**
> We have further investigated the client scalability (by incorporating more clients during training) of our method on the medical cancer diagnosis dataset (i.e., Camelyon17). Below, we present the average test performance by incorporating more clients into training. Our method shows a stable increasing trend compared with other methods.
> |     Method    | Client 1,2 | Client 1,2,3 | Client 1,2,3,4 | Client 1,2,3,4,5 |
> |:-------------:|:----------:|:------------:|:--------------:|:----------------:|
> |     FedAvg    |      96.56 |        95.61 |          95.63 |            95.00 |
> |     FedAlt    |      96.56 |        96.58 |          96.71 |            96.66 |
> |     FedBN     |      96.83 |        95.40 |          96.46 |            95.53 |
> | LG-Mix (Ours) |      98.22 |        98.49 |          98.58 |            98.75 |
>
>
> **(c) Explore applications in the new domain of brain-computer interface data:**
> Following your suggestion, we further explored the application on the brain-computer interface (BCI) data, which consists in classifying the mental imagery EEG (electroencephalography) datasets. The EEG data is a type of data collected from electrodes placed on the scalp to measure and record electrical activity in the brain. This data can provide valuable insights into brain function and can be used in various fields such as neuroscience, psychology, and medicine. Specifically, we use four datasets from the MOABB BCI benchmark [1], each of those EEG datasets have been acquired on different subjects and have different numbers of channels, resulting in heterogeneous features. We compared our method with all other methods in our manuscript and reported five evaluation metrics, including Accuracy, AUC, Sensitivity, Specificity, and F1-score. The results are shown in the following table. Our method achieves the highest performance on all metrics, with significant improvements of 2.73% accuracy by the second-best method. More details can be found in Section D.2 in the Appendix.
>
>
> **BCI dataset**
> |   Method  | Acuracy |  AUC  |  Sensitivity  |  Specificity  |   F1-score  |
> |:---------:|:-------:|:-----:|:-----:|:-----:|:-----:|
> |   FedAvg  |  67.38  | 69.71 | 68.14 | 66.77 | 67.29 |
> |   APFL    | *71.01*  | *75.48* | *71.21* | 71.08 | *70.97* |
> |   L2SGD   |  67.39  | 69.96 | 69.11 | 65.88 | 67.31 |
> |   FedAlt  |  68.46  | 72.74 | 69.72 | 67.52 | 68.45 |
> | PerFedAvg |  67.47  | 71.25 | 64.22 | 70.56 | 67.35 |
> |   FedBN   |  68.06  | 70.62 | 68.28 | 68.03 | 68.03 |
> |  FedFOMO  |  64.69  | 68.56 | 65.36 | 64.18 | 64.53 |
> |   FedRep  |  68.10  | 69.82 | 69.75 | 66.68 | 68.01 |
> |  FedBABU  |  68.03  | 70.45 | 69.99 | 66.25 | 67.95 |
> |   FedHKD  |  68.69  | 70.34 | 64.99 | *72.32* | 68.44 |
> |LG-Mix (Ours) |  **73.74**  | **78.11** | **75.19** | **72.58** | **73.72** |
>
>
> [1] Jayaram, Vinay, and Alexandre Barachant. "MOABB: trustworthy algorithm benchmarking for BCIs." Journal of neural engineering 15, no. 6 (2018): 066011.

---

> ### Author Response · Authors · 2023-11-21
> **Rebuttal by authors (2/3)**
>
> >- How is eq. 4 derived? please give more explanation on it. y is label, w(t) is the local model weights, what's the meaning of y-w(t)?
>
>
> Eq. 4 is derived by following Du et al.’s work [1], which uses the Gram matrix to characterize the optimization process of gradient descent for neural networks. Eq. 4 is the major theorem (Theorem 4.1) in Du et al.’s work to show the convergence rate of gradient descent. It measures the prediction error, which is the difference between the model prediction and the ground truth label.
>
>
> $y-w(t)$ should be $y-y(t)$ in Eq.(4), which denotes the prediction error. It is a typo, many thanks for pointing this out, and sorry for the confusion.
> In our paper, $w(t)$ is used to denote the local model weights, and $y(t)=f(w(t),x)$ denotes the model prediction with model weights $w(t)$ and $f: \mathbb{R}^d \rightarrow \mathbb{R}$ is the model function. We have revised our manuscript's Eq.(4) to avoid confusion.
>
> [1] Du, Simon S., Xiyu Zhai, Barnabas Poczos, and Aarti Singh. "Gradient descent provably optimizes over-parameterized neural networks." ICLR (2019).
>
>
> > - What's the cost (e.g., GPU memory cost, computational time) of the proposed method, compared to the baseline methods?
>
>
> The GPU memory cost and computational time cost of our method come from two aspects. First, we store a copy of the global model during local training. Second, our method calculates latent features and the trace of latent feature matrices.
>
>
> For GPU memory cost, we monitored the peak GPU memory cost of each method and listed the costs in the table below. Our method takes a reasonable cost compared with other baseline methods. Please note that we do not report FedRep, FedBABU, and FedHKD on the retinal dataset because they are specifically designed for classification tasks.
>
>
> **GPU memory cost (MB)**
> |     Method    | Digits5 | OfficeCaltech10 | DomainNet | Camelyon17 | Retinal |
> |:-------------:|:-------:|:---------------:|:---------:|:----------:|:-------:|
> |     FedAvg    |  270.24 |     1042.32     |  1042.32  |   1096.89  | 1814.29 |
> |     APFL      |  600.77 |     2993.48     |  2993.48  |   2955.24  | 4504.02 |
> |     L2SGD     |  456.18 |     2057.62     |  1157.62  |   2072.09  | 2329.91 |
> |     FedAlt    |  458.27 |     1042.33     |  1042.33  |   1096.89  | 2297.57 |
> |   PerFedAvg   |  270.24 |     1042.32     |  1042.32  |   1096.89  | 1814.29 |
> |     FedBN     |  458.27 |     1042.32     |  1042.32  |   1097.99  | 2295.21 |
> |    FedFOMO    |  458.28 |     1948.98     |  1948.98  |   1985.06  | 2837.97 |
> |     FedRep    |  458.27 |     1042.30     |  1042.30  |   1096.89  |    -    |
> |    FedBABU    |  458.27 |     1042.30     |  1042.30  |   1096.89  |    -    |
> |     FedHKD    |  864.28 |     3589.56     |  3589.56  |   3021.02  |    -    |
> | LG-Mix (Ours) |  472.37 |     2101.01     |  2101.01  |   1905.38  | 3730.69 |
>
>
> For the computational time, we counted the training time of all methods. The GPU type used for training is GeForce RTX 2080 Ti. From the results in the table, it can be observed that our method lies in a reasonable range with a slightly higher cost than FedAvg, which is a trade-off between computational efficiency and performance.
>
>
> **Training time (hour)**
> |     Method    | Digits5 | OfficeCaltech10 | DomainNet | Camelyon17 | Retinal |
> |:-------------:|:-------:|:---------------:|:---------:|:----------:|:-------:|
> |     FedAvg    |   3.70  |       1.28      |   17.37   |    55.01   |  25.95  |
> |     APFL      |   4.38  |       1.70      |   19.59   |    79.12   |  26.36  |
> |     L2SGD     |   3.83  |       1.40      |   18.42   |    57.17   |  26.79  |
> |     FedAlt    |   4.09  |       1.47      |   28.77   |    76.80   |  34.16  |
> |   PerFedAvg   |   3.79  |       1.45      |   18.66   |    56.97   |  26.44  |
> |     FedBN     |   3.73  |       1.26      |   17.43   |    55.03   |  26.56  |
> |    FedFOMO    |   6.02  |       2.80      |   19.04   |    76.80   |  36.90  |
> |     FedRep    |   4.32  |       1.41      |   19.18   |    55.77   |    -    |
> |    FedBABU    |   3.96  |       1.31      |   15.92   |    62.12   |    -    |
> |     FedHKD    |  22.43  |       2.64      |   37.33   |    68.71   |    -    |
> | LG-Mix (Ours) |   5.86  |       1.32      |   18.92   |    57.98   |  26.55  |
> ​​

---

> ### Author Response · Authors · 2023-11-21
> **Rebuttal by authors (3/3)**
>
> >- What are the possible limitations of the proposed method, and how can they be addressed in future work?
>
> Thanks for the insightful question. One potential limitation of our method is that we approximate the trace calculation by only using the last-layer latent feature.
> While this approximation is reasonable and reflects the convergence rate in principle, it may not be an accurate estimation when using large foundation models as the model backbone. The convergence behavior may differ due to the model's architecture and complexity. There is a risk of obtaining less precise estimations in these scenarios. To address this limitation, one potential approach is to consider the prediction errors of local and global models. By incorporating the model errors into the mixing process, we can obtain more comprehensive measurements for combining local and global updates. However, exploring the use of the foundation model in federated learning is still an open question. Explorations are needed in future work to validate the effectiveness of our method further. We have included a detailed discussion of our method's limitations and potential extensions in Section E in the Appendix.

---

### Author Response · Authors · 2023-11-21
**Rebuttal by authors**

We express our sincere appreciation to the reviewers for dedicating their valuable time and effort to reviewing our paper, as well as for providing us with constructive and supportive comments, highlighting that our “_motivation is sound and efficient_”, “_approach is original and innovative_”,“_clarity of the paper is excellent_”, and we “_compared with many other relevant methods_”.

This rebuttal clarifies i8zc's concerns on the application in other domains, efqn's questions on training time and training loss, and addresses all other minor questions and discussion points from all reviews.

In addition, we have uploaded a new version of the manuscript to incorporate the suggested revisions.  The revisions are highlighted with different colors in response to different reviews: pink to reviewer i8zc, green to review efqn, and blue to review Ho8d.

---

### Meta-Review · Area_Chair_Rzuv · 2023-12-07

**Metareview:**

This paper proposes a novel technique to address the challenge of data heterogeneity in personalized federated learning. The proposed method considers the different convergence rate from the Neural Tangent Kernel, quantifies the importance of local and global updates according to the contribution to the training process, and subsequently mix these updates to accelerate the local training. The authors have theoretically analyzed and experimentally evaluated their method on five datasets with heterogeneous data features in natural and medical images via the importance coefficient. Overall, this paper contributes to the development of more effective and efficient personalized federated learning methods.

Strengths:

The perspective of mixing updates to solve the personalized federated learning problem is novel and useful in the heterogeneous dataset. This paper proposes a good illustration of the local efficiency of adopting both local training and global training. Furthermore, to accelerate the local training, the global update is introduced as an auxiliary correction via the measurement of the Neural Tangent Kernel. Both experiments and theoretical analysis guarantee the efficiency of the proposed method.

Weaknesses:

More experimental setups and details including hyperparameter selections should be clearly stated in the main text. The test for the specific last-layer Gram trace criterion is not clear enough and easy to understand, which needs to be further improved.

After the author's response and discussion with reviewers, most concerns are well solved. Although a few minor issues exist in the current version, the positive contributions of this work still beat weaknesses. Therefore, I recommend acceptance. Furthermore, we recommend the authors incorporate the rebuttal to the final version.

**Justification For Why Not Higher Score:**

N/A

**Justification For Why Not Lower Score:**

N/A

---

### Decision · Program_Chairs · 2024-01-16

Accept (poster)